

# Coupling urban drainage-wastewater systems and electric smart grids during dry periods: A gain/loss framework using the relative economic value with ensemble flow forecasts to predict the switch between management objectives

Vianney Courdent[1,2], Morten Grum[1,†], Thomas Munk-Nielsen[1], Peter S. Mikkelsen[2]

[1] Krüger Veolia, Søborg, 2860, Denmark
[2] Department of Environmental Engineering, Technical University of Denmark, Kgs. Lyngby, 2800, Denmark
[†] Present address: Founder and CEO of WaterZerv, Environmental Services, Denmark.

*Correspondence to*: Vianney Courdent (vatc@env.dtu.dk)

**Abstract.** Precipitation is the major perturbation to the flow in urban drainage and wastewater systems. Flow forecast, generated by coupling rainfall predictions with a hydrologic runoff model, can potentially be used to optimise the operation of Integrated Urban Drainage-Wastewater Systems (IUDWS) during both wet and dry weather periods. Numerical Weather Prediction (NWP) models have significantly improved in recent years; increasing their spatial and temporal resolution. Finer resolution NWP are suitable for urban catchment scale applications, providing longer lead time than radar extrapolation.

However, forecasts are inevitably uncertain and fine resolution is especially challenging for NWP. This uncertainty is commonly addressed in meteorology with Ensemble Prediction Systems (EPS). Handling uncertainty is challenging for decision makers and hence tools are necessary to provide insight on ensemble forecast usage and to support the rationality of decisions (i.e. forecasts are uncertain therefore errors will be made, decision makers need tools to justify their choices, demonstrating that these choices are beneficial in the long run).

This study presents an economic framework to support the decision making process by providing information on when acting on the forecast is beneficial and how to handle the EPS. The Relative Economic Value (*REV*) approach associates economic values to the potential outcomes and determines the preferential use of the EPS forecast. The envelope curve of the *REV* diagram combines the results from each probability forecast to provide the highest relative economic value for a given gain-loss ratio. This approach is traditionally used at larger scales to assess mitigation measures for adverse events (i.e. the

actions are taken when events are forecasted). The specificity of this study is to optimise the energy consumption in IUDWS during low flow periods by exploiting the electrical smart grid market (i.e. the actions are taken when no events are forecasted). Furthermore, the results demonstrate the benefit of NWP neighbourhood post-processing methods to enhance the forecast skill and increase the range of beneficial use.






## 1 Introduction

The primary objective of combined urban drainage systems (UDS) and wastewater treatment plants (WWTP) is to convey and treat waste water and to prevent flooding and combined sewer overflows (CSO). In order to achieve these objectives pipes and detention basins in combined UDS are dimensioned to cope with relatively large rain events. Typically, surcharge of manholes and flooding is only allowed to occur on average every 10 years (as on the Danish regulations, (Harremoës et al., 2005)) whereas overflow occurs more frequently depending on the local environmental regulations, from e.g. 10 times per year to once in 10 years. This means that during dry weather the flow is relatively low compared with the conveyance capacity of the UDS and that the storage capacity is left unused. Rainfall only occurs rarely, on average 7.2 % of the time in the Damhuså catchment, Copenhagen (more details in Sect. 2.3.), which means that during dry weather, the management objectives of Integrated Urban Drainage-Wastewater Systems (IUDWS) could potentially be switched from the normal operational focus on CSO and flood prevention to other goals like minimization of energy consumption and reduction of $CO_2$ emissions.

Denmark has the political ambitions to have a fossil fuel free energy system by 2050 which requires the development of renewable energy sources (Ministry of Foreign Affairs of Denmark, 2016). A main critic towards renewables like wind and solar energy is their intermittent nature. Therefore a key parameter for the transition to a green energy system is the implementation of an electric Smart Grid with flexible, proactive consumers to balance the fluctuating power production (Hadjsaïd and Sabonnodiere, 2012). Energy markets are developed, as part of the Smart Grid, to align electricity production and consumption thought bids and offers. Hence the electricity price is based on supply and demand, creating an economic incentive to distribute the energy consumption in time (e.g. shifting non-essential energy consumption out of the consumption peaks). For further detailed history and description of electricity markets see (Weron, 2006).

IUDWS can potentially be used actively to take advantage of the energy market variation. Wastewater for example contains organic matter which can be converted to biogas at the WWTP, and the biogas production process may provide some energy storage that is potentially useful in a smart grid context. Furthermore, during dry periods, the unused storage in the UDS can be used as buffer to control the timing of the energy consumption associated with wastewater transportation and treatment.

Figure 1 highlights that both wastewater production and energy consumption are driven by human activities and therefore have similar daily pattern. This means that the energy is generally more expensive when the need for wastewater transportation and treatment is peaking. The energy market is also influence by other parameters (e.g. the solar and wind intensity) but on yearly average the impact of the daily consumption can be observed.

Aymerich et al. (2015) investigated the relation between the energy consumption and energy cost at a WWTP in regard to energy tariff structures (i.e. energy markets). The aeration process represents between 50 and 70 % of the WWTP process energy consumption (Rosso and Stenstrom, 2005). Leu et al. (2009) studied the impact of a varying wastewater load on the oxygen transfer efficiency and aeration costs considering to the daily variation of the power rates and showed a potential to reduce the average power costs, within the limitations of the WWTP storage capacity. Bjerg et al. (2015) investigated the use



of the storage volume in the pipe system upstream from a WWTP in Kolding (Denmark) to detain wastewater and utilise the energy price fluctuations. However, such optimization requires information on the incoming loads (i.e. flow predictions) to know when it is safe to optimize the energy consumption (i.e. when it is dry weather) and when to prepare and operate the IUWDS to cope with large inflows during wet weather. Such flow predictions should ideally cover the forecast horizon of

the smart grid market, i.e. 1 to 2 days (e.g. day-head energy market (Zugno, 2013)), which requires the use of Numerical Weather Prediction (NWP) models.

NWPs are already in use in other fields such as wind and solar power production prediction (Bacher et al., 2009; Giebel et al., 2005), streamflow forecasting (Cuo et al., 2011; Shrestha et al., 2013), reservoir inflow prediction (Collischonn et al., 2007), flood forecasting (Damrath et al., 2000) and typhoon forecasting (Chang et al., 2015). Uncertainty is a challenge for

NWPs especially for precipitation which is non-continuous and highly variable in both space and time. To tackle this problem meteorologists commonly generate Ensemble Prediction Systems (EPS) by perturbing the initial conditions and the physics of the NWP models to generate a number of ensemble members (EM) that represent an ensemble spread. The quality of an EPS can be quantitatively assessed based on various forecast characteristics. The relative operating characteristic (ROC) (Mason, 1982) is used to measure the discrimination skill (i.e. the ability to discriminate events and non-events) of an

EPS, by plotting the empirical Probability of Detection ($PoD$) versus the Probability of False Detection ($PoFD$). Using an EPS increases the event discrimination skill by providing a larger range of predictions than an individual deterministic forecast. Each fraction of ensemble members $f_{EM}$ from 1/N to 1 (with N the number of EMs) provides a set of prediction skills. Choosing a value of $f_{EM}$ converts the EPS to a single forecast which treats forecast with a probability greater than $f_{EM}$ as the occurring event. For example, for $f_{EM} = \frac{1}{N}$ then at least 1 EM must predict an event for the general prediction to be an

event, and for $f_{EM} = 1$ then all the EMs must predict an event for the general prediction to be an event (Courdent et al., 2016).

The development of high-resolution limited area NWP models have led to more realistic-appearing forecasts. Convective precipitations are described in an explicit and more detailed way using mesoscale atmospheric processes (Sun et al., 2014). These developments foster the opportunity of UDS applications which require fine temporal and spatial resolution. However

precipitation is one of the most difficult variables to forecast on an urban scale due to its large variability in space, time and intensity (Du, 2007). Precipitation forecast uncertainties increase rapidly with decreasing special grid size as inevitable errors in position and timing of rain cells are amplified with the increase in resolution. EPS aim at describing this uncertainty, but EPS are generally under dispersive and unable to capture all sources of uncertainty. NWP post-processing methods (also called pre-processing from a hydrological modelling point of view) are thus necessary to obtain reliable

probabilistic forecast as explained in (WWRP/WGNE 2009). Courdent et al. (2016) described NWP post-processing methods for urban drainage flow forecasting and compared their event discrimination skills. The neighbourhood methods (Theis et al., 2005) can e.g. be used to enhance the forecast skill by accounting for potentially misplaced rain events. The maximal threat method NWP post-processing, used in this study, considered the highest rainfall prediction within a given





area surrounding the catchment. The radius of the neighbouring area included is used as a parameter during the decision making, in addition to $f_{EM}$.

This article presents a framework for objectively optimizing EPS forecast-based decision making in management of IUDWS by selecting the decision threshold $f_{EM}$ and post-processing neighbourhood method with the highest Relative Economic

Value (*REV*) given the specific problem at hand; as example we consider the decision problem of switching from normal operation focusing on flow management to dry weather operation focusing on energy optimization linking with the Smart Grid. To measure the usefulness of weather forecasts, the forecast skills have to be converted to potential economic benefits for the user decision making process. Richardson (2000) used the *REV* to assess the economic benefit of road gritting to prevent the formation of ice using weather models in comparison to using purely climatological information (i.e. the

statistical behaviour of the weather, e.g. the return period of an event). Economical values were assigned to the different prediction outcomes described in a contingency table: (a) hits, (b) false alarms, (c) misses and (d) correct negatives. These economic values represent the benefit of taking actions (or non-action) when the forecast reveals to be correct against the drawbacks of those actions (or non-action) in case of forecast error.

EPS provides a range of prediction skills characterised by the combined choice of post-processing method and decision

threshold ($f_{EM}$) used to predict an event. The *REV* of each combination is quantified considering the occasions when the forecast proves to be beneficial, detrimental or neutral to the user and the economic value associated with these situations. The higher the cost of inappropriate action relative to the potential gain, the more certainty the user requires about the forecast before he/she takes action.

Previous studies on *REV* analysis typically assessed the benefit of prevention measures mitigating severe weather events,

such as frost (Richardson, 2000), intense precipitation (Atger, 2001), river floods (Roulin, 2006) and typhoons (Chang et al., 2015), expressed as a cost-loss ratio. This study develops a different perspective, assessing the potential benefit of optimizing IUDWS when the forecast predicts periods with low flow (i.e. dry weather when no events are forecasted). Therefore our decision model is not based on a cost-loss ratio but a gain-loss ratio. Furthermore the mentioned studies consider a fixed ratio whereas in our case, (i) the gain depends on smart grid variations and (ii) the loss is related to the risk

of Combined Sewer Overflow (CSO) and the negative impact on the WWTP operation. Hence the gain-loss ratio and the optimum combination of post-processing method and decision threshold need to be reassessed for each time step.

This paper is organized as follows: section 2 introduces the DMI-HIRLAM-S05 weather model used to produce the data used in our study, the used post-processing method and the hydrological rainfall-runoff model used. Section 3 describes the prediction performance evaluation methods used, including the ROC and the *REV* diagrams. Results and prediction

examples are presented and discussed in section 4 and finally, section 5 provides the conclusions.





## 2 Material: NWP data, study case and hydrological model

As emphasized by Shrestha et al. (2013)  the evaluation of NWP model precipitation forecasts for streamflow forecasting should be done with a hydrological perspective. Therefore as recommended by Pappenberger et al. (2008), the evaluation of urban drainage flow forecasts is in this paper based on a coupled meteorological and hydrological model. Hence, the forecast skills are assessed based on discharge predictions and discharge observations rather than precipitation forecasts and precipitation observations. This methodology considers the importance of the dominant hydrological processes and the non-linear error transformation by the hydrological model.

This section describes the Numerical Weather Prediction (NWP) model and data used in the study. Then the used of post-processing neighbourhood methods is presented, the urban catchment study case is presented and the hydrological model is described and finally, the energy marked data used is presented.

### 2.1 The EPS HIRLAM-DMI-S05 Numerical Weather Prediction (NWP) model

The rainfall forecasts used in this study were generated by the DMI-HIRLAM-S05 model and were provided by the Danish Meteorological Institute (DMI). This NWP model has a horizontal resolution of 0.05° (approx. 5.6 km) and a forecast horizon of 54 hours with hourly time step predictions. New forecast are generated every 6 hours at 0h, 6h, 12h and 18h. The DMI-HIRLAM-S05 ensemble is a 2-dimensional EPS comprising 25 members based on 5 different initial conditions and 5 different model structures. For further description of the processes and parameters mentioned above, see the HIRLAM technical documentation (Unden et al., 2002), the DMI technical report (Feddersen, 2009) and the HIRLAM website (http://www.hirlam.org/). The study uses 2 years of archived EPS NWP data (from June 2014 to May 2016).

### 2.2 Enhancing forecast by post-processing NWP EPS data

Two NWP post-processing methods were used in this study, (i) the realistic catchment "weighted areal overlap" method which only considers the grid cells overlapping the hydrologic catchment and weighs them based on the percentage of overlap and (ii) the "maximal threat" in the surroundings method which considers cells within a defined radius around the catchment. The maximal threat method combines the worst scenario approach and the neighbourhood method developed by Theis et al. (2005), and accounts for neighbourhood cells in the prediction as illustrated by Figure 2. Hence, the maximal threat approach considers as input, for each ensemble member, the highest rainfall intensity in the surroundings. This method keeps the same ensemble size as the weighted areal overlap method and reduces the number of missed events but increases the number of wrongly or over-predicted events (Courdent et al., 2016).

### 2.3 Study case and hydrological model description

The economic framework developed in this study was applied on the Damhuså urban drainage catchment (Copenhagen, Denmark). This highly urbanised area with compact residential housing is equipped with a combined sewer system which





conveys wastewater, rainfall runoff from paved surfaces and infiltration inflow especially in the winter months. This catchment was chosen for the absence of major flow control infrastructures affecting its hydraulic response in order to simplify the modelling approach needed for our demonstration.

Rainfall observation data were obtained from the national Danish SVK rain gauge network (blue circles on Figure 3 which is operated by the Danish Meteorological Institute (DMI) and the Water Pollution Committee of the Danish Society of Engineers (SVK – Spildevandskomiteen, in Danish). The rainfall measurements were recorded with a 1 minute temporal and a 0.2 mm volumetric resolution, for more information see (Jørgensen et al., 1998). The catchment outlet (red hexagon on Figure 3) is a combined sewer pipe interceptor with a maximum capacity of 10,000 $m^3$ $h^{-1}$. Once this threshold is reached CSOs occur. The overflowing water is discharged untreated into a nearby small river (Damhusåen) while the remaining flow is discharged through the interceptor pipe which is monitored using an electromagnetic flow meter with a 2 minutes temporal resolution and operated by the utility company HOFOR.

This study is based on event prediction by characterising the flow status in the IUDWS and distinguishing two domains, (i) periods with high flows during which the management objective is to maximize the hydraulic capacity of the WWTP to limit the impact of CSO etc. and (ii) periods with low flows during which the management objectives can be switched to WWTP operational efficiency minimizing energy consumption, etc. The event definition should be evaluated relatively to the specific IUDWS and low-flow optimisation scheme in focus. In this study the occurrence of an event is defined by a flow exceedance of 4,000 $m^3$ $h^{-1}$ over a 1 hour period. The occurrence (or non-occurrence) of a high flow event is assessed for each hourly time step forecasted (summing up to a total of n event assessments).

The hydrological model is composed of 3 main conceptual parts: (i) the wastewater flow from households is modelled using $2^{nd}$-order Fourier series, see e.g. (Langergraber et al., 2008) calibrated to dry weather flow observations (Fig. 1), (ii) the fast rainfall-runoff from impervious areas is represented by a lumped conceptual model using the Nash linear reservoir cascade concept (Nash, 1957) and (iii) the slow runoff (caused e.g. by infiltration-inflow) is also modelled based on the Nash linear reservoir cascade concept using a wetness index characterised by the monthly potential evaporation and previous rainfall events, see further details in (Courdent et al., 2016).

## 2.4. Energy market data

Power systems are experiencing considerable changes, facing a paradigm shift from a centralized network delivering one-directional electricity to a distributed network with active consumers. Indeed, new energy sources, e.g. wind, solar, biomass and cogeneration plants (combined heat power), are spatially highly distributed. Furthermore, renewable energy sources (wind and solar power) are characterized by high temporal variability and uncertainty over production levels which have significant impact on other sources to protect the integrity of the network (Hadjsaïd and Sabonnodiere, 2012). The electric smart grid aims at facing these challenges. The European Technology Platform for Smart Grids defines the concept of Smart Grids as an "*electricity network that can intelligently integrate the actions of all users connected to it - generators, consumers and those that do both - in order to efficiently deliver sustainable, economic and secure electricity supplies*"



([www.smartgrids.eu/](www.smartgrids.eu/)). This study used historical market data from the energy market provider Nord Pool. Energy prices are defined per hours over a geographical area, the geographical area corresponding to our case study is DK2 which covers the entire Zealand ([http://www.nordpoolspot.com/](http://www.nordpoolspot.com/)).

## 3 Methodology

The methods used in this study are mainly borrowed from the field of meteorology and further developed to match the needs in IUDWS management. This section is divided in three parts. First the contingency table, *PoD* and *PoFD* are introduced and the Brier Skill Score (*BSS*) is presented. Then the ROC diagram which displays the discrimination skill of an EPS is explained. Finally the Relative Economic Value method which translates *PoD* and *PoFD* of an EPS into an economic value of the forecast is explained and adapted from a cost/loss to a gain/loss ratio basis.

### 3.1 Contingency table

The probability that the flow will exceed a given threshold is estimated as the fraction of EMs predicting an event. The ensemble (probability) forecast can be converted to a single binary forecast by selecting a decision threshold ($f_{EM}$) (threshold probability). If the fraction of ensemble members (EM) predicting an event is higher or equal to the decision threshold ($f_{EM}$) then an event is forecasted.

The empirical performance over a period of time of a binary forecast can be summarized in a 2x2 contingency table showing the number of correctly and incorrectly forecasted events occurring or not occurring (Table 1). Hits (*a*) represent the correct positives, false alarms (*b*) represent the false positives, misses (*c*) represent the false negatives and the correct negatives (*d*) represent the correct forecasts of no events occurring.

Measures of performance of a sequence of binary forecasts can be formulated as a function of these four outcomes (*a*, *b*, *c*

and *d*), which sum up to *n*.

Table 2 displays the verification measures used in this paper, a comprehensive review and further description of verification measures can be found in the meteorological literature, e.g. (WWRP/WGNE, 2009) and (Wilks, 2011).

The probability of detection (*PoD*) is defined as the fraction of occurrences of events that were correctly forecasted (i.e. hits), while the probability of false detection (*PoFD*) is the fraction of non-occurrences of events that were incorrectly

forecasted (i.e. false alarms). The empirical occurrence frequency ($\mu$) expresses climatological information about the occurrence of events.

### 3.2 Brier skill score

The Brier Score (Brier, 1950) assesses forecast quality of discrete probability forecasts predicting binary outcomes (i.e. "events" and "non-events") and is comparable to the mean square error. For a given $t^{th}$ hourly forecast time step, the





forecasted probability of an event ($0 \leq f_{EM,t} \leq 1$) is compared to the observation ($y_t$). If the $t^{th}$ observation is an event (or non-event) then $y_t = 1$ (or $y_t = 0$).

$$BS = \frac{1}{n} \sum_{t=1}^{n} (f_{EM,t} - y_t)^2 \qquad (1)$$

The Brier Skill Score (*BSS*) is formulated as a skill score related to a reference forecast, e.g. climatology in meteorology. In our case the reference forecast is based on the frequency of occurrence of events during the recorded forecast period ($\mu$). A positive value of the *BSS* indicates that forecast is beneficial compared to the reference forecast.

$$BSS = 1 - \frac{BS}{BS_{ref}} \quad \text{with} \quad BS_{ref} = \frac{1}{n} \sum_{t=1}^{n} (\mu - y_t)^2 \qquad (2)$$

### 3.3 Relative Operating Characteristic (ROC)

The Relative Operating Characteristic (ROC), which originates from signal detection theory (Mason, 1982), measures the discrimination ability (i.e. the ability to discriminate events and non-events) of an EPS. The ROC plots the *PoD* versus the *PoFD* using a set of decreasing probability decision thresholds (Figure 4). The selection of a lower decision threshold $f_{EM}$ to convert the ensemble forecast to a single forecast is more conservative towards correctly predicting events. Therefore the *PoD* will be higher but the *PoFD* will increase as well.

The ROC diagram of the flow domain distinction using the weighted areal overlap NWP post-processing method is displayed on Figure 4. The blue dots represent the discrimination skill of each individual ensemble member. Figure 4 shows that all ensemble members have comparable discrimination skill. The red dots correspond to the discrimination skills from all decision thresholds, from $f_{EM} = 1$ at the bottom left (i.e. all ensemble member should agree on the event occurrence) to $f_{EM} = \frac{1}{N}$ on the top right (i.e. the prediction of an event from a single ensemble member is enough to consider an occurrence). Figure 4 underlines that EPS and decision thresholds provide a larger range of available prediction skills than the ensemble member individually. The choice of a decision threshold represents a trade-off between predicting events correctly and generating false alarms.

The skill score of a ROC diagram is calculated based on the area under the curve (*ROCA*). The *ROCA* ranges from 0 to 1, a score of 1 corresponds to a perfect forecast and a score of 0.5 corresponds to the skill of a random forecast based on the probability of occurrence ($\mu$).

### 3.4 Relative Economic Value (REV)

A proper evaluation of the benefits of a forecast system should not only consider the forecasts skill e.g. using *PoD* and *PoFD*, or *BSS*. A detailed knowledge of the decision-making process is needed to answer the question: "how does this skill translate to an economic value of a forecast?". Furthermore when using ensemble forecasts the following question should be





answered as well: "which decision threshold and NWP post-processing method for the EPS is the most beneficial for my purpose?".

The economic benefit from a forecast depends on the alternative courses of action and their consequences. Each course of action is associated with a cost and leads to economic benefit or loss depending on the observed outcome. The task is thus to choose the appropriate actions which will maximize the expected gain or minimize the expected loss. The usefulness of the forecast can thus be quantified by considering the occasions when the forecast was beneficial, detrimental or neutral with respect to the process of decision making.

The relative economic value of our urban hydrological prediction system is here inspired by on the relatively simple cost-loss ratio decision model introduced by (Richardson, 2000). Richardson developed this approach to assess the economic value of taking costly actions to mitigate the consequences of forecasted adverse weather events in order to reduce the potential loss associated with them. The decision threshold that can empirically be shown to lead to the lowest expense on the long term should be adopted. Richardson illustrated his approach for the problem of roads gritting to prevent the formation of ice. Subsequently Roulin (2006) used this approach to investigate the benefit of river flow mitigation measures for two catchments in Belgium and (Chang et al., 2015) applied it to assess the relevance of typhoon mitigation measures in Taiwan.

All these studies consider adverse events which can be mitigated at a cost, reducing the loss associated these events, and their decision models are therefore based on a cost-loss ratio. This study investigates a different perspective. Instead of taking mitigating measures when adverse events are predicted, the system is optimized when no events are predicted in order to achieve a positive gain, and left under its traditional management when event are predicted. Therefore our decision model is based on a gain-loss ratio. During low flow periods, when no events are forecasted, the management objective is switched to energy consumption by utilising the Smart Grid energy market leading to a gain (G). As a consequence miss predicted high flow events will jeopardise the IUDWS, e.g. the detention basins may not be empty in time. These negative outcomes are represented by a loss (L). In case of forecasted events (hits and false alarms) the management objectives of the IUDWS remain unchanged. The economic outcome of these two situations remains the same therefore a null value is assigned to them, Table 3.

Furthermore Richardson (2000) used a static ratio, the cost of mitigation measures and reduction of loss associated were fixed. This study encompasses the possibility of dynamic, time dependent gain-loss ratio. Indeed, the gain (G) from switching the management objectives to energy optimisation depends on the state of the energy market at the given time. Similarly the loss (L) resulting from miss predicted events is related to the current status of the IUWDS, e.g. the volume of water stored.

Based on

Table 2 and Table 3 the expected economic value of using the forecast for decision making over one time step (n represents the total number of time steps) can be expressed empirically as:





$$E_{forecast} = \frac{d*G - c*L}{n} \tag{3}$$

In case of a perfect forecast ($b = c = 0$) the economic value would be:

$$E_{perfect} = d * \frac{G}{n} = (1 - \mu) * G \tag{4}$$

If no forecasts are available, the optimal course of action can be determined based on the empirical frequency of occurrence

of an event, $\mu$ (climatological information in case of weather event as for (Richardson, 2000)). The two possible courses of action are either to always optimize the system despite the losses or to never optimize the system. $E_{statistic}$ considers the highest economic value between these two courses of action (Eq. 4); never optimizing (i.e. the IUDWS management is unchanged) would lead to an null economic value whereas always optimizing would lead to a gain G associated to a loss L when events do occur.

$$E_{statistic} = \max(G - \mu * L, 0) \tag{5}$$

The Relative Economic Value (*REV*), as defined by (Richardson, 2000), compares the benefit of acting on a given forecast to the benefit which would be achieved by acting on a perfect forecast as a ratio (Eq. 5).

$$REV = \frac{E_{forecast} - E_{statistic}}{E_{perfect} - E_{statisic}} \tag{6}$$

The *REV* expressed by Eq. 6 can be reformulated using Eq. 3, Eq. 4 and Eq. 5 and expressed as a function of the *PoD*, the

*PoFD*, the frequency of occurrence ($\mu$) and the gain-loss ratio ( $\alpha = \frac{G}{L}$ ) as shown by Eq. 7 and displayed on Figure 5.

$$REV = \frac{\alpha*(1-\mu)*(1-PoFD) - (1-PoD)*\mu - \max(\alpha-\mu,0)}{\alpha*(1-\mu) - \max(\alpha-\mu,0)} \tag{7}$$

The possible value of the *REV* ranges from 1, corresponding to a perfect forecast, to minus infinity. In case of positive *REV* the use of the forecast is beneficial, whereas a negative *REV* indicates that using statistical information and either always or never optimising the IUDWS yields a better economic value than using the weather forecast. Hence the *REV* can be divided

in 3 domains: (i) the interval on the right of the curve in which it is preferable to always optimise (dotted domain on the right side of Figure 5), (ii) the interval with positive *REV* covered by the curve in which using the forecast in beneficial (middle domain on Figure 5)  and (iii) the interval on the left in which it is preferable to never optimise (crosshatched domain on the left side of Figure 5). Assuming that a perfect knowledge of the future yields a benefit β (compared to purely statistical information), then using the actual forecast provides a benefit to the user of  $(100 * REV)$ % of β.

Figure 6 displays the ROC diagram and the *REV-α* relationship for flow forecast based on the catchment weighted areal overlap post-processing method. As explained in Sect 3.3. the ROC diagram describes the EPS forecast discrimination skill for the different decision thresholds, $f_{EM}$. To support decision making the ROC diagram is converted to the *REV-α* relationship. Each point of the ROC diagram (Figure 6a) represents a discrimination skill (*PoD*, *PoFD*) for a given decision





threshold based on the fraction of ensemble members predicting an event ($f_{EM}$). For each of these points the *REV* can be determined as a function of the gain-loss ratio α (Eq. 7 and Figure 5). The curves on Figure 6b show the *REV*-α relationship for the decision thresholds ($f_{EM}$) highlighted in Fig. 6a. The *REV* is thus closely related to the ROC as indicated by (Richardson, 2000; Zhu et al., 2002).

The green dot (legend 5) on Figure 6a corresponds to a decision threshold $f_{EM} = 1/25$ and provides the highest *PoD* for this EPS; the *REV* associated with it, i.e. the green line (legend 5) on Figure 6b, leads to the highest *REV* for low α values (below 0.105) which corresponds to high a negative impact of missed events. Other decision thresholds yield better *REV* for higher *α*, e.g. the decision threshold $f_{EM} = 5/25$ corresponding to the red dot (legend 4) on Figure 6a provides the highest *REV* (legend 4) for *α* within the range [0.16; 0.18]. Hence as demonstrated by Richardson (2000) the ensemble has better

discrimination and can provide higher *REV* to a wider range of users (i.e. larger interval with positive *REV*) than any individual deterministic forecast (colour line) as illustrated by the envelope curve.

## 4 Results and discussion

### 4.1 ROC, REV and NWP post-processing methods.

The implementation of the IUDWS energy consumption optimisation scheme is challenged by potentially missed high flow

events. Indeed, these situations would lead to inappropriate management, jeopardizing the performance of the IUDWS. As explained in Sect. 2.2, post-processing methods can be applied to enhance the NWP, e.g. by accounting for potentially misplaced events which can have significant impact at an urban hydrology scale. Figure 7 displays the result considering the NWP maximal threat post-processing EPS method with a 6 grid cells radius around the catchment. This approach is more conservative towards avoiding missed events and yields higher *PoD* at the cost of higher *PoFD*, which extends the ROC

diagram. The ROC curves on Figure 7a show that the two approaches are complementary; the areal overlap method provides better discrimination skill for low *PoFD* whereas the maximal threat EPS post-processing method provides better discrimination skill for higher *PoFD*. The ROCA of each approach is respectively 0.86 and 0.91 and the ROCA merging both approaches is 0.92.

This new ROC curve results in the extension of the α-interval with positive *REV* which characterises the range of beneficial

forecast use (Figure 7b). To ease the comparison the area under the envelop curve of the areal overlap approach is displayed in grey colour as background on Figure 7b, and Table 4 gives intervals of positive α for both approaches. The weighted areal overlap provides a slightly better upper bound whereas the maximal threat approach significantly expands the interval of positive *REV* for low α values. Therefore, using this NWP post-processing approach increases the range of beneficial forecast usages.

The comparison between these two NWP-post processing approaches using the Brier Skill Score (BSS) shows a deterioration of the forecast skill when using the maximal threat approach, which has a negative *BSS* indicating that the forecast performs worse that the reference forecast based on the frequency of occurrence of an event ($\mu$). This decrease in





performance can be explained by an increase in false alarms due to the precautions towards not missing a major rain event of this approach. This result underlines the need an economical assessment rather than purely forecast skills to draw conclusions of the usefulness of a forecast for a given decision making situation.

## 4.2 Examples of EPS flow domain prediction

In order to illustrate the different situations of decision making taken as a starting point for this paper (i.e. when to switch from flow management to energy management and vice versa) a range of 4 theoretical α-values were considered, Table 5. The two outer α-values yield negative *REV* indicating that using the forecast data is not beneficial in these cases. The two other *α*-values yield positive *REV* indicating that using the forecast is beneficial in these cases. The decision threshold ($f_{EM}$) generating the highest relative benefit based on empirical data are displayed on Figure 8 and in Table 5.

The coupled hydro-meteorological model provides an ensemble prediction of the flow at the catchment outlet for the incoming 2 days. Figure 9 provides an example of prediction. The first panel, Figure 9a, displays the energy market during those two days, providing insight in the variation of the energy price and the $CO_2$ footprint through the proportion of wind energy. The shown data are based on historical values but similar information are forecasted by the electric smart grid and available in real time. The fluctuation of the energy market for both parameters (Figure 9a) illustrates the variation of the α-

value in relation to the potential gain during a given period. During the first day (29[th] April 2015) the energy price ranges from 24 € to 32 € and the proportion of wind energy varies from 15 % to 49 %, whereas during the second day (30[th] April 2015) the energy price range from 23 € to 41 € and the proportion of wind energy varies from 1 % to above 53 %. Hence the switch of consumption of 1 MWh can yield up to 8 € during the first day and up to 18 € during the second day. Furthermore energy consumption optimisation based on economic objectives will also yield environmental benefits and vice versa. Indeed

the energy price and the proportion of wind energy are negatively correlated. The North Pool Energy Market DK2 covering the Copenhagen area e.g. has a Pearson correlation coefficient of -0.52 between energy price and proportion of wind energy in 2015, indicating a moderate negative linear relationship.

Figure 9b represents the flow forecast based on the catchment weighted areal overlap approach and Figure 9c represents the flow forecast based on the maximal threat EPS approach with a 6 grid cells radius. The measured flow during this period

shows two minor rain events without significant flow impact the first day and a major rain event leading to high flows exceeding the 4,000 m[3] in the IUDWS the second day. Figure 9b illustrates the difficulty of the prediction to have a correct timing, most EMs predict the high flow event but often too early. It can be noticed that due to the conservativeness of this second approach the EPS plume of flow forecasts overestimates the observed flow (in red), which explains the worsing of the *BSS* when using this approach.

The best flow domain predictions, considering a given *α*, is provided by the decision threshold defined using the *REV* method presented in Sect. 3.4. As displayed in Table 5, the highest *REV* for *α*=1/20 (respectively *α*=1/100) is achieved using the NWP post-processing approach "Maximal Threat EPS" with $f_{EM} = \frac{11}{25}$ (respectively $f_{EM} = \frac{1}{25}$). The flow domain





predictions based on these criteria and on the EPS flow forecast displayed on Figure 9b and Figure 9c are shown by the blue hatched (respectively plain blue) colour in Figure 9d.

## 4.3 Outlooks

The predictions and therefore the skills of the EPS are based on a coupled meteorological and hydraulic model. This study used a lumped conceptual hydraulic model; a more detailed hydrological model including stochastic processes and on-line assimilation of flow measurements might improve the prediction and thereby improve the *REV* further. Similarly, NWP models are continuously improving and benefit from the constant increase of computational calculation power to enhance their resolution and ensemble size. The techniques of data assimilation from radar measurement into NWP models are also consistently improving (Korsholm et al., 2015). Weather services are collaborating to continuously improve their meteorological models. The HIRLAM consortium which developed the model structure of the DMI-HIRLAM-S05 NWP used in this study is e.g. currently developing and launching the non-hydrostatic convection-permitting HARMONIE model in cooperation with Météo-France and ALADIN, and EPS with forecast horizons of up to two weeks are also available at the European level (http://www.ecmwf.int/). Therefore the accuracy and lead time of the prediction and hence the potential benefit from the framework developed in this article is expected to increase in the future.

Additionally, other characteristic of NWP can be utilised. The DMI-HIRLAM-S05 model e.g. generates a new 54 h EPS forecast every 6 hours, and thereby the successive forecasts are overlapping each other. The forecast consistency, or in reverse the "forecast jump", provides valuable information on forecast uncertainty which could be utilised in the decision making process. For example the time-lagged method (Mittermaier, 2007) uses consecutive forecast overlapping to extend the EPS and enhance the predictions (i.e. the horizon of the forecast is reduced but its ensemble size is increased). This may increase the range of positive *REV* and allow use of the concept for decisions related to other problems than the energy optimization problem studied here.

Control systems can be decomposed into different layers in a hierarchy. Mollerup et al. (2016) presents a methodological approach to the design of optimising control strategies for sewer systems. The framework presented in this paper targets the upper layer of the hierarchy presented by Mollerup et al. (2016): the management of objectives where switching between different operational modes may take place. Completely different optimising control strategies, including model predictive control techniques, may then run under different operation conditions – such as the "flow control" and "energy optimisation" operational modes considered in this paper. Implementing such a switching system in practice requires that the gain/loss ratio expressing the economic consequences associated with the outcomes of the different course of actions used for the *REV* is quantified, which requires further research on monetization of non-market goods, e.g. $CO_2$ footprint or the environmental impact of CSOs, and may depend on local circumstances.



## 5 Conclusion

An ensemble flow prediction system for an IUDWS was developed using the DMI-HIRLAM-S05 EPS as input to a hydrological model. This system was tested on an urban catchment in the Copenhagen area based on recorded rainfall forecasts and flow data for the period from June 2014 to May 2016. Ensemble forecasting requires adapting the management rules in order to use probability forecasts instead of a deterministic forecast. The usefulness of the forecast should be evaluated not only based on its quality in terms of traditional skill scores but also based on its economic value for the daily decision-making process of the forecast user considered. The decision problem considered here is the switching from normal flow management during high flow periods (wet weather) to Smart Grid energy optimization during low flow periods (dry weather).

This article presents a framework to support decision making based on the prediction of the occurrence or no occurrence of an event using an ensemble predicting system (EPS). The outcomes (gain for positives and loss for negatives) of the different possible courses of action are valued to determine the Relative Economical Value (*REV*) of using the forecast. The *REV* is closely related to the ROC diagram, which assesses the range of discrimination skills of an ensemble forecast. Hence a *REV* curve, as a function of the gain/loss ratio $\alpha$, can be generated for each probability threshold ($f_{EM}$) of the EPS. This method was developed in order to switch the IUDWS management objective from flow management to energy optimization utilising the electric Smart Grid when low flow periods are predicted. This approach is based on daily optimisation when non-events (dry weather) are forecasted and differs from previous studies based in the *REV* concept which investigated mitigation measures taking place when adverse events are forecasted (e.g. flood, tornado) using a cost/loss ratio. In our approach for a given gain-loss ratio $\alpha$, the probability threshold ($f_{EM}$) corresponding to the highest *REV*, symbolised by the envelop curve, should be applied to maximize the benefit of the optimisation scheme. If the gain-loss ratio is outside the range of positive *REV*, then using the forecast is not beneficial. The gain-loss ratio $\alpha$ is dynamic and depends e.g. on the potential gain from utilising the variation of the Smart Grid energy market at a given point in time.

Two NWP post-processing methods were tested: (i) a realistic approach based on the weighted areal overlap between the NWP grid cells and the hydrological catchment and (ii) a more conservative approach considering the maximal rainfall threat in the catchment vicinity. The second approach leads to a deterioration of classic forecast validation scores such as Brier Skill Score due to a significant increase in the number of false alarms. However, this approach proves to be beneficial in regard to the decision making process especially when considering a low gain-loss ratio $\alpha$ for which missed forecasted events are highly detrimental. Indeed, the maximal threat NWP neighbourhood post-processing method improves the range of discrimination skill of the predictions shown on ROC diagram and therefore provides a larger range of positive *REV*, increasing the range of beneficial forecast usage. This is underling the importance of assessing the forecast usefulness based on its potential economic value rather than solely on the usual forecast skills.





## Acknowledgment

This research was financially supported by the industrial PhD programme of the Innovation Fund Denmark. The catchment and flow data were kindly provided by Copenhagen Utility Company (HOFOR). We would like to thank the Danish Meteorological Institute (DMI), especially Henrik Feddersen, for providing EPS data from their NWP model DMI-
HIRLAM-S05.

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

**Table 1: Contingency table (with n the sample size)**

| Event forecasted | Event observed | | |
|---|---|---|---|
| | Yes | No | |
| Yes | hits ($a$) | false alarms ($b$) | $a + b$ |
| No | misses ($c$) | correct negatives ($d$) | $c + d$ |
| | $a + c$ | $b + d$ | $a + b + c + d = n$ |



**Table 2: Verification measures based on the contingency table**

| Score | Formula | Range | Perfect |
|---|---|---|---|
| Probability of detection, *PoD* | $a/(a+c)$ | [0,1] | 1 |
| Probability of false detection, *PoFD* | $b/(b+d)$ | [0,1] | 0 |
| Occurrence frequency of events, $\mu$ | $(a+c)/n$ | [0,1] | na |

**Table 3 : Economical value assigned to the different outcomes of the contingency table (L:loss; G:gain).**

| | Event observed | |
|---|---|---|
| **Event forecasted** | Yes | No |
| Yes | 0 | 0 |
| No | L | G |

**Table 4: *BSS* and *REV* characteristics for the two different NWP post-processing methods**

| | *ROCA* | $\alpha$-interval | | *BBS* |
|---|---|---|---|---|
| | | Lower bound | Upper bound | |
| Weighted areal overlap | 0.86 | 0.0208 | 0.3955 | 0.14 |
| Maximal threat 6 cells radius | 0.91 | 0.0049 | 0.3940 | -1.52 |

**Table 5: Decision threshold and *REV* for the theoretical 4 $\alpha$-values considered, using the maximal threat post-processing method.**

| $\alpha$ | *REV* | Prediction criteria | |
|---|---|---|---|
| | | Decision Threshold | NWP post-processing |
| 1/2 | Negative | Always optimise | |
| 1/20 | 0.59 | $f_{EM} = 11/25$ | Maximal Treat EPS |
| 1/100 | 0.30 | $f_{EM} = 1/25$ | Maximal Treat EPS |
| 1/500 | Negative | Never optimise | |





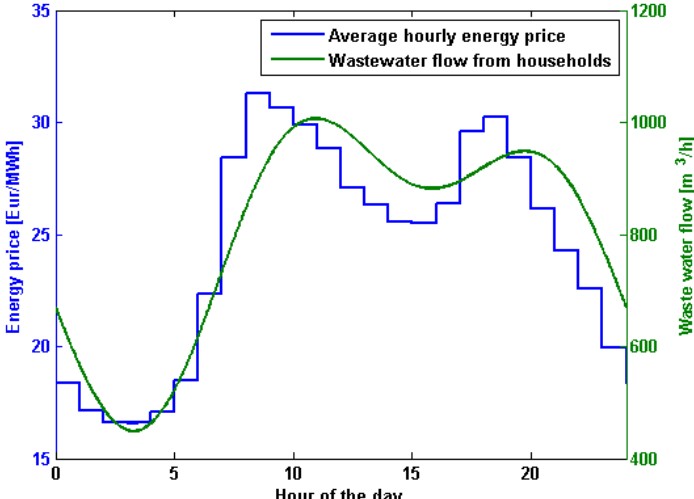

**Figure 1: Yearly average (2015) of hourly energy price for the energy market DK2 covering the Copenhagen region (in blue, data from http://www.nordpoolspot.com/). Calibrated daily variation of the dry weather flow for the Damhusåen catchment (green) used for demonstration in this paper see further details in Sect. 2.3.**

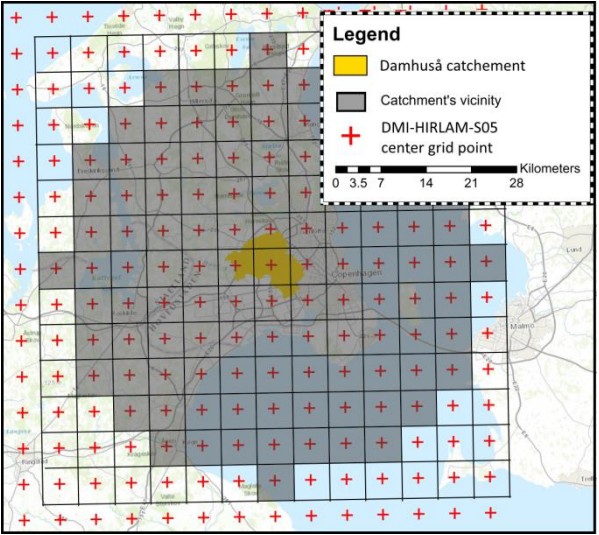

**Figure 2: Illustration of the 6 grid cells radius used by the maximal threat neighbourhood approach, for the Damhusåen catchment used for demonstration in this paper (Courdent et al., 2016).**





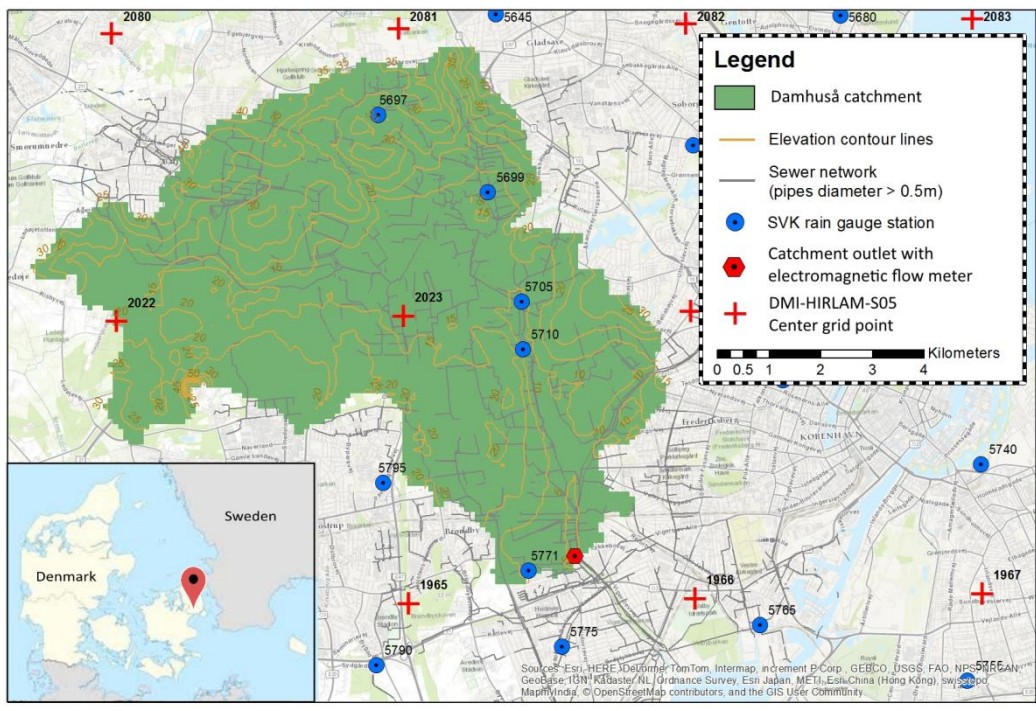

**Figure 3: The Damhuså urban drainage catchment, Copenhagen, Denmark (contributing area, green area on the map, 67 km2).**

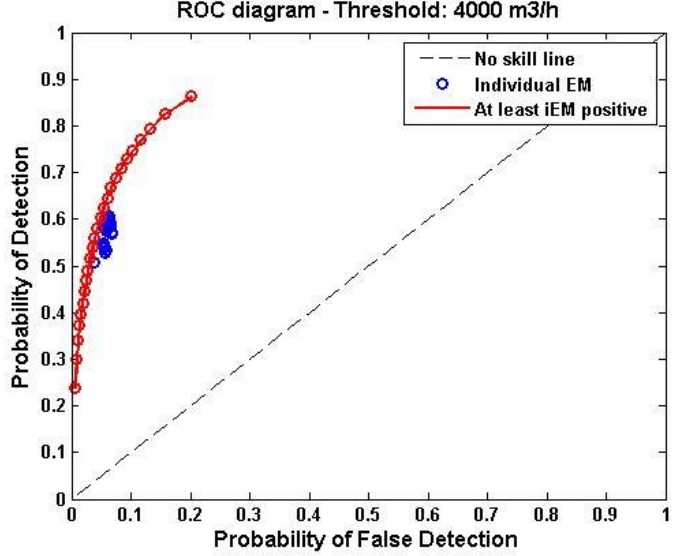

**Figure 4: Example of Relative Operating Characteristic (ROC) diagram.**





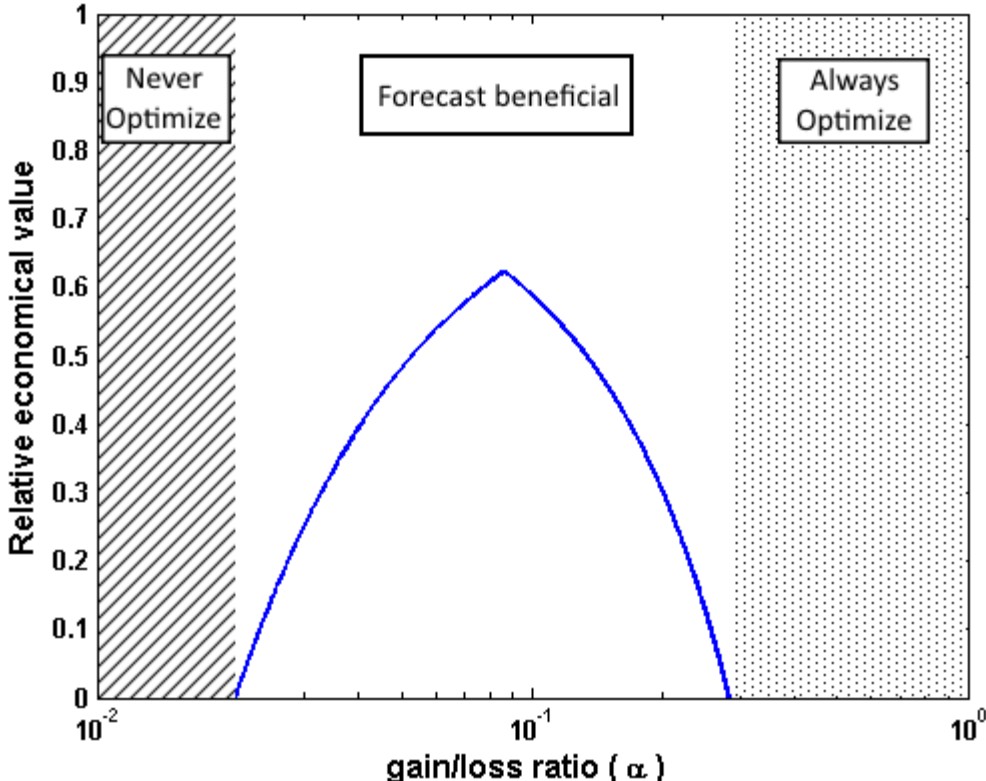

**Figure 5: The 3 domains of operation of the *REV* curve as a function of the gain-loss ratio *α*.**





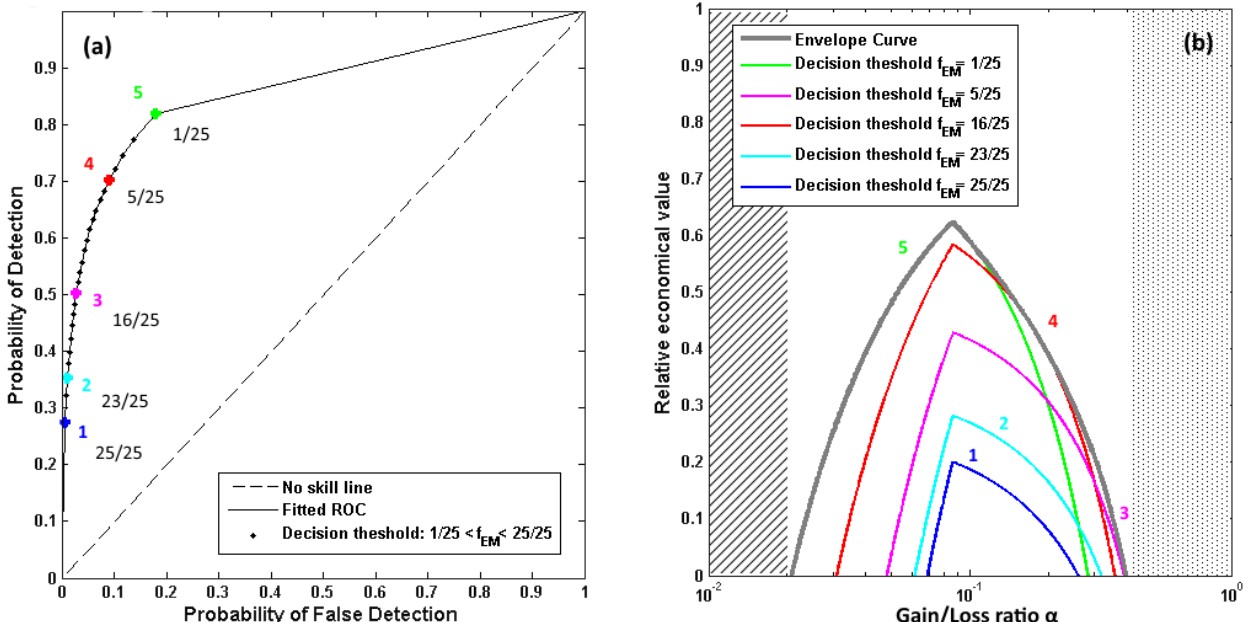

**Figure 6: ROC and REV diagram for flow domain forecast based on catchment weighted areal overlap.**

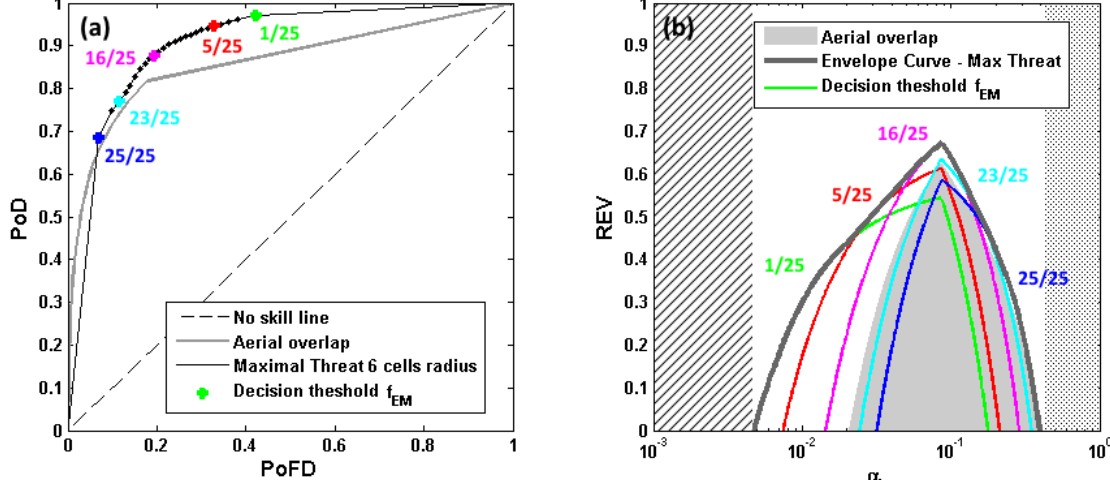

5    **Figure 7: ROC and REV diagram for flow forecasts considering the two NWP post-processing methods: The maximal treat EPS method with a neighbourhood radius of 6 grid cells in colour and the catchment weighted areal overlap method in grey colour as background.**




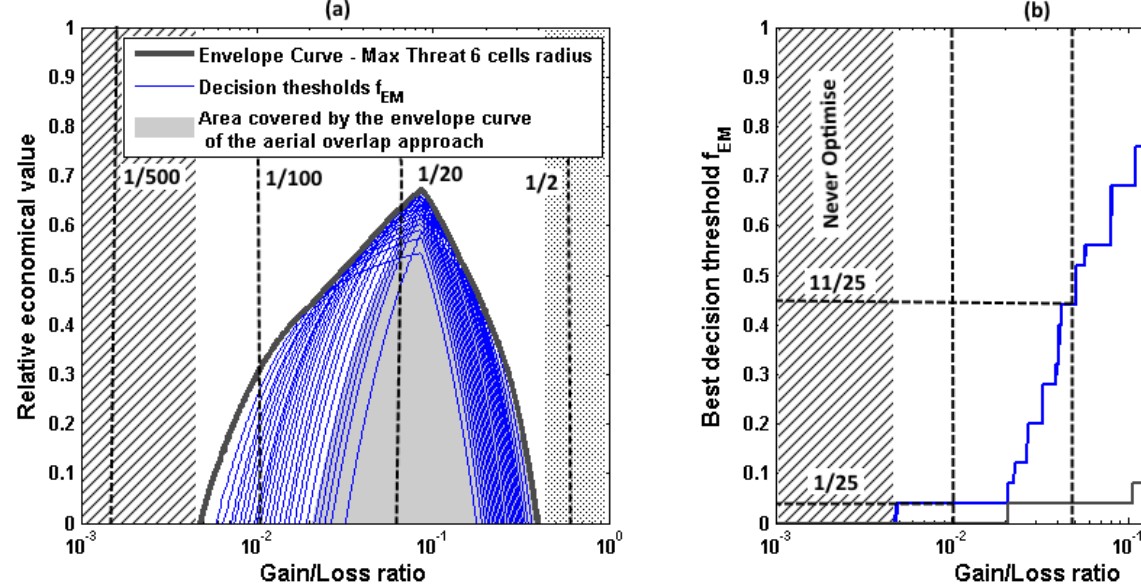

**Figure 8:** *REV* **curves for the EPS NWP post-processing maximal threat in a radius of 6 grid cells from the catchment (a, left plot) and best decision threshold according to the α-value (b, right plot), in blue for the maximal threshold approach and in grey for the areal overlap approach.**





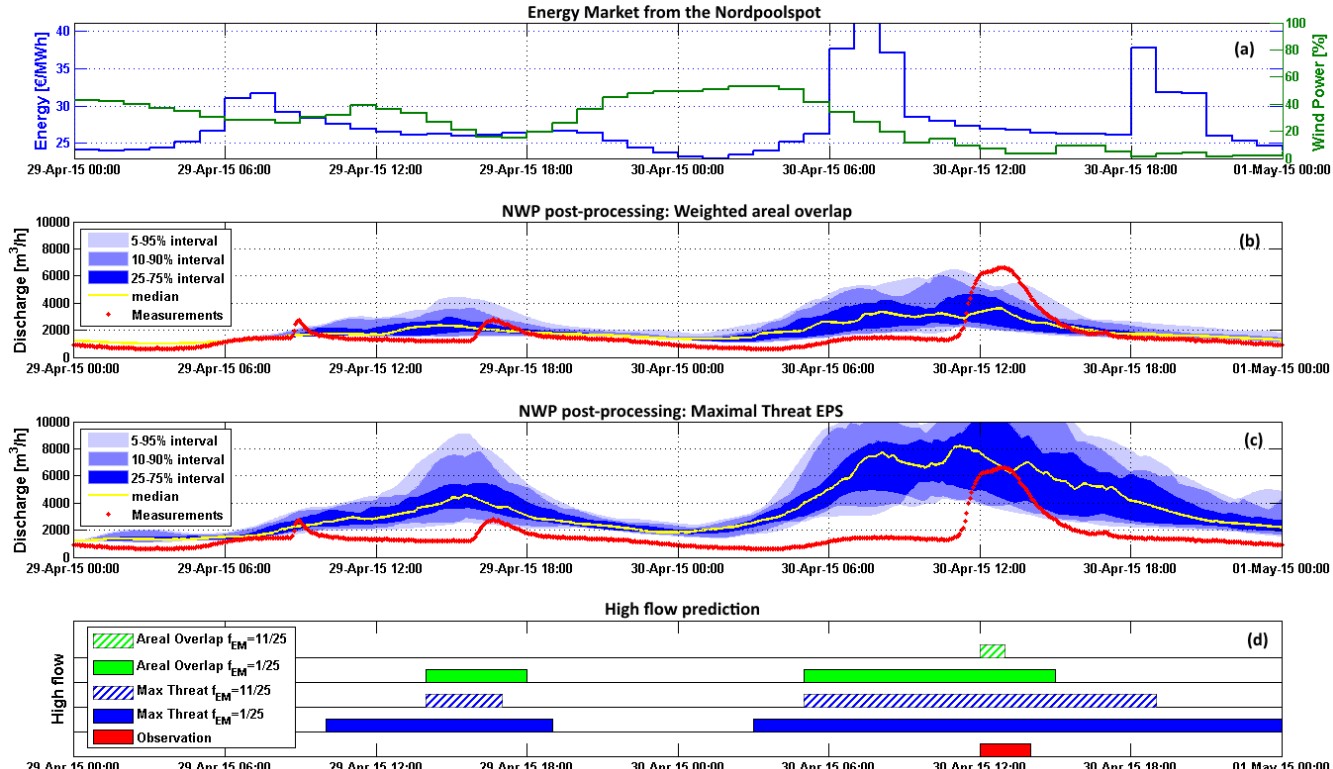

**Figure 9: Example illustration of the EPS flow prediction system for two selected days, 29-30 April 2015. Energy market parameters, energy price and proportion of wind power (1, top plot), ensemble flow predictions using the areal average (b) and maximal threat (c) post-processing methods, and (d) flow domain predictions for the two post-processing methods and for each two decisions thresholds, cf. Table 5 (coloured areas imply that an event is predicted, otherwise not).**