# Peer review of "A gain-loss framework based on ensemble flow forecasts to switch the urban drainage-wastewater system management towards energy optimization during dry periods."

_Hydrology and Earth System Sciences, 2016_

## Referee Comment (RC1) · Anonymous Referee #1 · 4 Nov 2016

Review comments on manuscript hess-2016-522 Coupling urban drainage-wastewater systems and electric smart grids during dry periods: A gain/loss framework using the relative economic value with ensemble flow forecasts to predict the switch between management objectives

General comments

The manuscript presents an interesting topic within the scope of the journal. While

enough information on the used methodology is given to enable a rough understanding, the reader will not be able to reproduce the applied methods based on that information. Result interpretation will be difficult for the reader without more specific information on the used hydrological model (its calibration and accuracy) and the WWTP and the potential impact of the described method on its performance in terms of energy consumption and effluent water quality. More information in the results section on the performance of the proposed method and a discussion of its risks and benefits would be very desirable as without them it is difficult to draw clear conclusions about the value of the method. For these reasons, I suggest to accept the paper for a major revision thereby also giving the authors the opportunity to address a number of structural deficiencies and language related details.

Specific comments

P1 L1: This is one long title. Would "Coupling urban drainage-wastewater systems and electric smart grids during dry periods" not suffice? Even then I find the title somewhat misleading: Is the WWTP not continuously coupled to the smart grid? If I understand correctly, rather than coupling and uncoupling, only the temporal amount of energy consumption is optimized depending on the hydraulic condition.

P2 L8ff: Maybe I misunderstand this sentence, but it seems to suggest that only because rain only occurs 7 % of the time it makes sense to look into energy optimization. Would it make less sense if you had 10 % of rain? Could you clarify/rephrase this?

P2 L8ff: Dry weather flow rarely can be defined by "no rain", usually (sufficient sewer network size and event intensity) wet weather conditions will be predominant for several hours after a rain event has ended (as the proposed method does by using the flow rather than the rain as a trigger for switching between control objectives). I suggest rephrasing this section accordingly.

P2 L10ff (same phrase again, sorry): Striving for energy optimization and emission reduction of WWTP is standard practice for many years – both during dry and wet

weather. With this background in mind it would be better to slightly rephrase and cite some relevant literature here to avoid this phrase being interpreted as a novel suggestion as such.

P3 L15ff: Some details are discussed here that re-occur in section 3. Delete here?

P4 L5: A clear(er) definition of REV (as eg in the abstract) would be critical here as it forms the basis of the paper

P5 L20ff: Are methods (i) and (ii) defined in literature? They seem to be a mix of other methods. It would not be possible to reproduce your method from this section. Please give more specific information on the used methods for enhancing the forecast.

P6 section 2 and especially 2.2 discusses methods. I'd suggest to move these parts to section 3

P6 L20ff: You mention that the DWF module of the model is calibrated, but there are no details on the calibration of the Nash cascades. These details seem essential for accurate predictions, please add them to this section.

P5, section 2.3: No details on the WWTP and its energy consumption are given while this forms the focus of the paper. Changed flow regimes at the influent will necessarily cause changes in energy consumption of the WWTP. How are these considered in this study?

P6 L31ff: Move to introduction?

P7 L5ff: Leave out? Does not add anything to the understanding of the reader.

P11 L5ff: are these not results that should be moved to the next section?

Figure 8 is not interpreted in the text. Omit?

P12 L13: How big are the uncertainties in the energy price forecasts? Could you comment on their (potential) influence on your method?

P12 L15ff: In this scenario, it is possible to save 26 €MWh during 2 days. I suggest to add information that answers at least a number of the following questions in order for the reader to be able to understand the meaning of this result: Is this a representative result? What is the average/peak energy consumption of the WWTP? What is the maximum amount of energy that could be switched? How much could be saved during the 2 years of data you used? How does that impact the total energy consumption of the WWTP ('switching' could result in an increase as well as a decrease)? What is the influence on the WWTP effluent performance and emission of greenhouse gasses?

P12 L15ff: What is the cost of the suggested system (at the least the NWP data will have to be purchased + some man-hours for keeping the real-time system up and running) as compared to its benefits? It seems these considerations should be included in order to judge the actual gains produced by the system.

P12 L24 "optimization based on economic objectives will also yield environmental benefits": This seems a much too broad statement that should be explained or based on a citation. In the context of this paper, it seems that it would be perfectly possible to create a scenario where 'switching' energy consumption would lead to an overall increase of energy consumption (e.g. by running the blowers on a frequency at which they are less efficient than when not 'switching'), but a decrease in cost.

P12 L26: "most EMs predict the high flow event but often too early" Is this a problem of the precipitation prediction or the hydrological model? This would be difficult for the reader to judge (without information on the calibration of the hydrological model). Maybe you could add a line in figure 9: the output of the hydrological model given the observed rain. This would also (more or less) address the above comment for P6 L20ff.

Technical corrections

P2 L18: "... consumption thought bids" –> "... consumption through bids"

P2 L27 "... is also influence by..." –> "... is also influenced by..."

P3 L1 "... detain ..." → "...store..."

P3 L22 "The development of ... models have led..." –> "The development of ... models has led..."

P3 L26 "...decreasing special grid size..." –> "...decreasing spatial grid size..."

P3 L27 "...under dispersive..." –> "...underdispersive..."

P4 L8: "...the used of post-processing ..." –> "...the use of post-processing ..."

P4 L10: "...energy marked ..." –> "...energy market ..."

P6 L4: "...circles of Figure 3 which..." –> "...circles of Figure 3), which..."

P8 L18:"...member..." –> "...members..."

P9 L8: "...on ..." –> omit

P9 L16: "...loss associated these events..." –> "...loss associated with these events..."

P9 L19 "...event..." –> "...events..."

P10 L21: "...the forecast in beneficial..." –> "...the forecast is beneficial..."

P11 L7: "...high a negative impact..." –> "...a high negative impact..."

P12 L2: "...the need an economical..." –> "...the need for an economical..."

Figure 6: In 6a 4 corresponds to red and 5/25. 6b: 4 is red and 16/25. Is this correct? What is "legend 5" in Figure 6a?

Figure 9, uppermost left ordinate: should this not be "energy price"?

P14 L30 "This is underling..." –> "This underlines..."?
* * *

---

## Referee Comment (RC2) · Anonymous Referee #2 · 8 Nov 2016

This paper presents an interesting water management situation where an economic framework is used and takes advantage of probabilistic predictions. The decision to be taken is to switch the management objectives from one side maximizing the hydraulic capacity of a wastewater treatment plant and limit the impact of combined sewer overflow to the other side minimizing the cost of energy consumption by controlling the timing of wastewater transportation and treatment taking the energy market into account. The problem is complex and tackled with appropriate tools and data. This manuscript

is worth being published in the journal conditioned on clarifications requested in the following.

The problem is exposed as a dynamic one (e.g. P4 L25-26, P9 L27-30, P14 L21-22). However it is solved as a static one: the relative economic value is presented as a function of the gain/loss ratio. The decision threshold (i.e. the fraction of members of the ensemble of predicted discharge exceeding a discharge threshold) beyond which the manager decides to switch from energy optimization to safety of the system is deduced from the envelope of separate curves. As for the choice of a method for the post-processing of numerical weather prediction model the maximum threat method extends significantly the range towards low gain/loss ratios resulting in positive relative economic values compared with the aerial overlap method. Having these results, the methodology has still to be proven in real dynamic situation i.e. where the decision to be taken at a given time depends on the decisions already taken. What is missing in order to that? An order of magnitude of the losses in case of combined sewer overflow, a hydraulic model able to reflect the management actions? The authors are asked to make clear the scope of the paper and either add new results or add comments in the outlooks.

I missed information regarding the methodology. No lead time is specified with the results. Are all the ensembles (2 years x 4 issue hours) used at hourly time step to the forecast horizon of 54 hours (P5 L14-18)? Or 2 days (P3 L5, P12 L11-26)? How the scores are computed regarding both the issue time and the lead time? In case the forecast horizon is 2 days, how do the authors deal with the decreasing skill scores or relative economic value of the predictions with the lead time?

Specific comments

P3 L26 "spatial" instead of "special". Do you have a reference for this assertion?

P4 L1 "The radius of the neighbouring area included is used as a parameter during the decision making, in addition to fEM." should move from introduction to the methodology

section (2 Material: … 2.2). P11 L18, Figure 7 Results considering a radius of 6 grid cells are presented. This radius has been optimized on REV? What is the sensitivity to this parameter?

P5 L14 UTC

P6 L18 and other occurrences of "forecasted" should be "forecast". "summing up to a total of n event assessments" : this part of the description methodology should be made much clearer (see general comments).

P7 L4 "Methodology" : section 2 involves also description of the methodology. Section 3 is more related with validation.

P10 L6 and other occurrences (e.g. Table 5, Figure 5, 8) "always optimize", "never optimize" sounds strange because the paper is all about optimization. Find a short reference to the two objectives (like "always energy objective").

P11 L20-29, Figure 7b, Table 4 The slightly better upper bound provided by the areal overlap method can't be seen on the figure. How the complementarity of both post-processing approaches can be used in a real situation? Through the gain/loss ration and the decision threshold?

P12 L15-18 "can yield up to 8 $€^{'}$ what is the order of energy consumption we are dealing with? What is missing is an overall estimation of the cost of energy during the 2 years and how much is gained during the same period using the switching strategy optimized based on REV results.

P24 Figure 9(a) Add a legend for the two curves.

---

## Referee Comment (RC3) · Anonymous Referee #3 · 30 Nov 2016

This is an interesting and useful paper investigating how the decision to switch the operational goal of a combined sewer system from flow objectives to energy/cost objectives based on rainfall forecasts should be made.

I can find little to fault the paper. My main query would be that I could find no details or reference as to how the system is operated during the 'optimised' phase apart from relatively vague statements. While the paper is interesting without this, it does limit the understanding / reproducibility. I can see that the omission of this may be due to

commercial reasons given the authors' affiliations, but it would be useful to explicitly state this.

It would also be useful to understand more details of the calibration of the hydrological model even if just a few short sentences.

While clearly outside the scope of this paper, it would be of great interest to see any results from a real world implementation of the proposed framework should it be implemented!

There are a number of minor corrections needing attention as listed below:

Page (P) 1, Line (L) 10: Last word should be 'forecasts'.

P1 L25: 'forecasted' should be 'forecast'.

P5, L8-9: I think 'Then the used of postprocessing neighbourhood methods is presented ...' would read better as 'Then the postprocessing neighbourhood methods are presented ...'

P6, L15: I think there should be a comma after 'operational efficiency'.

P9, L21: 'miss predicted' should be 'mis-predicted'.

P9, L31: shouldn't have a new line after 'Based on'.

P11, L32: should read '...forecast performs worse than the...'.

P12, L28: 'worsing' should be 'worsening'?

P20 Figure 3 caption: the 2 in km2 should be super-script.

P20 Figure 4: the 3 in m3/h should be super-script if possible.

P22 Figure 7 a and b: legend entry 'Aerial' should be 'Areal'.

P22 Figure 7 caption: 'treat' should be 'threat'.

Finally a very minor point, but the authors switch between the US-English spelling 'optimize' and the UK-English 'optimise' which should be easy make consistent.

—————————————————————

---

## Author Comment (AC1) · 17 Jan 2017

Dear reviewer,

We greatly appreciate the review and acknowledge that the comments and suggestions will lead to an improved paper. Our reply to the general comments:

The manuscript presents an interesting topic within the scope of the journal. While enough information on the used methodology is given to enable a rough understanding,

the reader will not be able to reproduce the applied methods based on that information. Result interpretation will be difficult for the reader without more specific information on the used hydrological model (its calibration and accuracy) [1] and the WWTP and the potential impact of the described method on its performance in terms of energy consumption and effluent water quality [2].

[1] Section 2.3 (P5 and 6) will be split in 2, the first on the study case which will be expended with more data on the WWTP and the second on the hydrological model which will be further developed. We will also explain with more clarity that further information on the hydrological model is available in (Courdent et al., 2016). As suggested in the specific comment 19, the model output based on rain gauge data input will be added on figure 9 to illustrate the performance of the hydrological model.

Courdent, V., Grum, M. and Mikkelsen, P. S.: Distinguishing high and low flow domains in urban drainage systems 2 days 25 ahead using numerical weather prediction ensembles, J. Hydrol., doi:http://dx.doi.org/10.1016/j.jhydrol.2016.08.015, 2016.

[2] Further information on the WWTP energy consumption will be added to the manuscript (e.g. the energy consumption per m3 treated in 2015 was 0.267 kWh/m3). Additional references on the proportion of the energy consumption related to the inflow/load will be added. E.g. the aeration of the bioreactor represents between 50 and 70

We do not have precise data regarding the impact on the effluent water quality. However the effluent water quality is of primary interest. The Danish WWTPs pay taxes on their effluent pollution load. Therefore the boundaries of the energy optimization scheme will be defined to respect the performance of the WWTP.

Aymerich, I., Rieger, L., Sobhani, R., Rosso, D. and Corominas, L.: The difference between energy consumption and energy cost: Modelling energy tariff structures for water resource recovery facilities, Water Res., 81, 113–123, doi:10.1016/j.watres.2015.04.033, 2015.

More information in the results section on the performance of the proposed method and a discussion of its risks and benefits would be very desirable as without them it is difficult to draw clear conclusions about the value of the method [3].

[3] The scope of this article is to develop a method to determine when the IUDWS management can be switched to optimizing its energy consumption, which is possible during dry weather (flow below a given threshold). Further information on the energy optimization scheme will be added, but for a comprehensive description of the optimization scheme we refer to a manuscript by R. Halvgaard et al. that is currently under review (see below). The framework developed in this article provides a framework to activate this optimisation scheme given the potential gain expected.

We agree that further information on results and performance would be appreciated and we are currently working towards it. Indeed, two large pipes will be constructed just before the inlet to the Damhusåen WWTP with the primary purpose to reduce CSO to cope with new regulations. These 2 pipes can contain a volume corresponding to one day of dry weather flow and would nicely fit the concept developed in this paper and in the paper by R. Halvgaard et al.

R. Halvgaard, L. Vezzaro, P. S. Mikkelsen, M. Grum, T. Munk-Nielsen, P. Tychsen, H. Madsen: Integrated Model Predictive Control of Wastewater Treatment Plants and Sewer Systems in a Smart Grid (In Review Process).

Please find appended to this reply our point to point responses to the received comments displayed as a Table in pdf format. We will make changes to the paper that accommodate the technical comments by the reviewers, including careful proofreading.

We would like to express our sincere thanks to the reviewers for their constructive comments and identification of areas in the manuscript which needed clarification.

On behalf of all the authors,

[Figure]

Vianney Courdent

Please also note the supplement to this comment:
http://www.hydrol-earth-syst-sci-discuss.net/hess-2016-522/hess-2016-522-AC1-supplement.pdf

**Supplement:**

Reply to the specific comments of reviewer 1.

| Referee # 1 | |
|---|---|
| Specific comments | |
| 1. P1 L1: This is one long title. Would "Coupling urban drainage-wastewater systems and electric smart grids during dry periods" not suffice? Even then I find the title somewhat misleading: Is the WWTP not continuously coupled to the smart grid? If I understand correctly, rather than coupling and uncoupling, only the temporal amount of energy consumption is optimized depending on the hydraulic condition. | We agree that it is a long title, as we aimed for clarity. We are suggesting the follow reduced title:

A gain-loss framework based on ensemble flow forecasts to switch the urban drainage-wastewater system management towards energy optimization during dry periods. |
| 2. P2 L8ff: Maybe I misunderstand this sentence, but it seems to suggest that only because rain only occurs 7 % of the time it makes sense to look into energy optimization. Would it make less sense if you had 10 % of rain? Could you clarify/rephrase this? | Indeed, an average occurrence of rain of 7% or 10% does not make much difference in regards of energy optimisation. We calculated the rain occurrence on our catchment to give an order of magnitude. We will rephrase the sentence for more clarity. |
| 3. P2 L8ff: Dry weather flow rarely can be defined by "no rain", usually (sufficient sewer network size and event intensity) wet weather conditions will be predominant for several hours after a rain event has ended (as the proposed method does by using the flow rather than the rain as a trigger for switching between control objectives). I suggest rephrasing this section accordingly. | Yes, we will rephrase the sentence for more clarity. |
| 4. P2 L10ff (same phrase again, sorry): Striving for energy optimization and emission reduction of WWTP is standard practice for many years – both during dry and wet weather. With this background in mind it would be better to slightly rephrase and cite some relevant literature here to avoid this phrase being interpreted as a novel suggestion as such. | Yes, striving for energy optimization and emission reduction of WWTP is standard practice for many years – both during dry and wet weather.

Our point was that Urban Drainage Systems (UDS) and the WWTP can be considered as an integrated system (IUDWS), using their interaction to facilitate an optimal operation of the entire system. Using the upstream system (UDS) as a buffer to control the energy consumption when possible.

We will add references and rephrase the sentence for more clarity. |
| 5. P3 L15ff: Some details are discussed here that re-occur in section 3. Delete here? | This part of the introduction describes the fraction of ensemble member ($f_{EM}$) which is also described in section 3.3. As suggested this part will be deleted to avoid reoccurrence. |
| 6. P4 L5: A clear(er) definition of REV (as eg in the abstract) would be critical here as it forms the basis of | We will further describe the REV in this part of the introduction for more clarity. |

| the paper | |
|---|---|
| 7. P5 L20ff: Are methods (i) and (ii) defined in literature? They seem to be a mix of other methods. It would not be possible to reproduce your method from this section. Please give more specific information on the used methods for enhancing the forecast. | We agree that the description of these methods is succinct, and we will make clearer that they are further elaborated in our just published manuscript (Courdent et al. 2016).

Courdent, V., Grum, M. and Mikkelsen, P. S.: Distinguishing high and low flow domains in urban drainage systems 2 days ahead using numerical weather prediction ensembles, J. Hydrol., doi:http://dx.doi.org/10.1016/j.jhydrol.2016.08.015, 2016. |
| 8. P6 section 2 and especially 2.2 discusses methods. I'd suggest to move these parts to section 3 | The NWP post-processing methods described in section 2.2 are developed in the previous manuscript (Courdent et al. 2016) and are used to generate input data to the model for this article.

It was decided to develop it under the data section to distinguish it from the core methods of this paper; the section will be modified to make this clearer. |
| 9. P6 L20ff: You mention that the DWF module of the model is calibrated, but there are no details on the calibration of the Nash cascades. These details seem essential for accurate predictions, please add them to this section. | As mentioned in the general comments [1] this section will be split in 2 to further describe the hydrological model, and the link to the previous article (Courdent et al. 2016) developing the hydrological model will be made clearer. |
| 10. P5, section 2.3: No details on the WWTP and its energy consumption are given while this forms the focus of the paper [1]. Changed flow regimes at the influent will necessarily cause changes in energy consumption of the WWTP. How are these considered in this study? [2] | [1]As mentioned in the reply to general comments, section 2.3 will be split in 2 and further detailsd will be given on the WWTP in the study case section.

[2] Indeed, the control of the energy consumption based on the energy market can result in a decrease of the cost together with an increase of the energy consumption. References will be added to underline this possibility, e.g.: (Aymerich et al., 2015).

This aspect needs to be addressed in a paper detailing the energy consumption optimization, which is out of the scope of this article.

Aymerich, I., Rieger, L., Sobhani, R., Rosso, D., Corominas, L., 2015. The difference between energy consumption and energy cost: Modelling energy tariff structures for water resource recovery facilities. Water |

| | Res. 81, 113–123. doi:10.1016/j.watres.2015.04.033 |
|---|---|
| 11. P6 L31ff: Move to introduction? | This part of the section 2.4 on energy market section will be moved to the introduction. |
| 12. P7 L5ff: Leave out? Does not add anything to the understanding of the reader. | The introduction to the methodology section will be removed. |
| 13. P11 L5ff: are these not results that should be moved to the next section? | The last part of section 3.3 describes the figure 6 to illustrate the method.

This part will be rephrased the underline the explanations on the method rather than the results. |
| 14. Figure 8 is not interpreted in the text. Omit? | Figure 8 is mentioned in P12 L9 and will be further interpreted in this paragraph. |
| 15. P12 L13: How big are the uncertainties in the energy price forecasts? Could you comment on their (potential) influence on your method? | Sorry, the term "forecasted" was inappropriate and will be changed. The energy price for the incoming day is set through the energy market (Nord pool for Denmark) based on bids and offers and is therefore fixed without uncertainty.

"Buyers and suppliers submit bids and offers for each hour of the next day and each hourly MCP (market clearing price) is set such that it balances supply and demand." (Weron 2006)

The smart grid section will be reshaped, part of it will be moved to the introduction (see reply to comment 11) and additional information on the electricity market will be added (there are different electricity markets with different lead times, e.g. the day-ahead market have 24 hours lead time whereas the intraday market has 1 hour lead time. Bids and offers made on the first market can be adapted on the second).

Weron, R.: Modeling and forecasting electricity loads and prices: A statistical approach, First Edit., John Wiley & Sons Ltd., 2006. |
| 16. P12 L15ff: In this scenario, it is possible to save 26 €MWh during 2 days. I suggest to add information that answers at least a number of the following questions in order for the reader to be able to understand the meaning of this result: Is this a representative result? What is the average/peak energy consumption of the WWTP? What is the maximum amount of energy that | This comment is similar to general comments [2].

Further information on the WWTP energy consumption (e.g. energy consumption per $m^3$) will be added to allow the reader to have a better understanding of the meaning of this result. |

| | |
|---|---|
| could be switched? How much could be saved during the 2 years of data you used? How does that impact the total energy consumption of the WWTP ('switching' could result in an increase as well as a decrease)? What is the influence on the WWTP effluent performance and emission of greenhouse gasses? | As mentioned in the reply to comment 10 and 18, energy optimization based on the energy price can result in an increase of the total energy consumption. References will be added to underline this possibility.

The energy consumption optimisation scheme (not developed in this article) has to include the WWTP performance within its decision criteria. E.g. (R. Halvgaard et al.) used the nitrogen concentration as a measure of effluent quality.

The impact on emission of greenhouse gasses was not directly assessed. However, daily peaks in waste water usually coincide with peak demand on the power grid, thus coinciding with the highest energy price periods. Hence, reducing these wastewater inflow peaks when energy cost are high will also benefit the energy system by reducing grid load and GHG emissions (due to the need for more carbon- intensive energy sources during peak power demand periods). |
| 17. P12 L15ff: What is the cost of the suggested system (at the least the NWP data will have to be purchased + some man-hours for keeping the real-time system up and running) as compared to its benefits? It seems these considerations should be included in order to judge the actual gains produced by the system. | We have some experiences from an implementation of this concept at the WWTP of Kolding, Denmark (125.000 PE), and we will add our main findings to answer this question. |
| 18. P12 L24 "optimization based on economic objectives will also yield environmental benefits": This seems a much too broad statement that should be explained or based on a citation. In the context of this paper, it seems that it would be perfectly possible to create a scenario where 'switching' energy consumption would lead to an overall increase of energy consumption (e.g. by running the blowers on a frequency at which they are less efficient than when not 'switching'), but a decrease in cost. | The intended message of this sentence is that the correlation between energy price and proportion of wind energy leads to the consumption of energy with a lower $CO_2$ footprint.
But indeed as you rightfully pointed out the optimisation can result in an increase of the overall energy consumption.

The sentence will be motived to clarify this point. |
| 19. P12 L26: "most EMs predict the high flow event but often too early" Is this a problem of the precipitation prediction or the hydrological model? This would be difficult for the reader to judge (without information on the calibration of the hydrological model). Maybe you could add a line in figure 9: the output of the hydrological model given the observed rain. This would also (more or less) address the above comment for P6 L20ff. | Thank you for the suggestion, the output from the hydrological model given the observed rain will be added to the figure 9. |

|  |  |
|---|---|
| Technical corrections | Thank you for the technical corrections, which will be accommodated in the revised version of the manuscript. |

---

## Author Comment (AC2) · 17 Jan 2017

Dear reviewer,

We greatly appreciate the review and acknowledge that the comments and suggestions will lead to an improved paper. Our reply to the general comments:

The problem is exposed as a dynamic one (e.g. P4 L25-26, P9 L27-30, P14 L21-22).However it is solved as a static one: the relative economic value is presented as a

function of the gain/loss ratio [1]. The decision threshold (i.e. the fraction of members of the ensemble of predicted discharge exceeding a discharge threshold) beyond which the manager decides to switch from energy optimization to safety of the system is deduced from the envelope of separate curves. As for the choice of a method for the post-processing of numerical weather prediction model the maximum threat method extends significantly the range towards low gain/loss ratios resulting in positive relative economic values compared with the aerial overlap method. Having these results, the methodology has still to be proven in real dynamic situation i.e. where the decision to be taken at a given time depends on the decisions already taken. What is missing in order to that? An order of magnitude of the losses in case of combined sewer overflow, a hydraulic model able to reflect the management actions? The authors are asked to make clear the scope of the paper and either add new results or add comments in the outlooks [2].

[1] Thank you to point this out, you are right, we need to clarify the use of the term "dynamic". The referenced articles on the REV ((Richardson, 2000; Roulin, 2006) are using fixed alpha values (the cost of mitigation measures and their benefits are assumed fixed). In our case the alpha value varies with time, the gain depends directly on the variation of the energy market. Hence the incentive to optimize the IUDWS for energy consumption changes with time. In this sense the alpha ratio is dynamic. But, indeed for at a given time, for given NWP forecast the alpha is fixed and the problem solved as a static one. We will make this distinction clearer.

[2] We agree that further information on results and performance would be appreciated and we are working towards it. Indeed, two large pipes will be constructed just before the inlet to the Damhusåen WWTP with the primary purpose to reduce CSO to cope with the new regulation. Those 2 pipes can contain a volume corresponding to one day of dry weather flow and would nicely fit the concept developed in this paper and in the paper by R. Halvgaard et al. R. Halvgaard, L. Vezzaro, P. S. Mikkelsen, M. Grum, T. Munk-Nielsen, P. Tychsen, H. Madsen: Integrated Model Predictive Control of

Wastewater Treatment Plants and Sewer Systems in a Smart Grid (In Review Process).

I missed information regarding the methodology. No lead time is specified with the results. Are all the ensembles (2 years x 4 issue hours) used at hourly time step to the forecast horizon of 54 hours (P5 L14-18)? Or 2 days (P3 L5, P12 L11-26)? How the scores are computed regarding both the issue time and the lead time? In case the forecast horizon is 2 days, how do the authors deal with the decreasing skill scores or relative economic value of the predictions with the lead time? [3]

[3] Indeed a decrease in the performance skill is observed with increasing lead time as was documented in (Courdent, 2016). The bids and offers on the daily energy market are made up to 36 hours in advance. Therefore, the different lead times are aggregated in the results. We will clarify this point in the methodology section.

Please find appended to this reply our point to point responses to the received comments displayed as a Table in pdf format. We will make changes to the paper that accommodate the technical comments by the reviewer, including careful proofreading. We would like to express our sincere thanks to the reviewers for their constructive comments and identification of areas in the manuscript which needed clarification.

On behalf of all the authors,

Vianney Courdent

———————————————

[Figure]

Reply to the specific comments of reviewer 2.

| Referee # 2 | |
|---|---|
| Specific comments | |
| 1. P3 L26 "spatial" instead of "special". Do you have a reference for this assertion? | Indeed it is a typo, we meant "spatial". |
| 2. P4 L1 "The radius of the neighbouring area included is used as a parameter during the decision making, in addition to fEM." should move from introduction to the methodology section (2 Material: : : : 2.2). P11 L18, Figure 7 Results considering a radius of 6 grid cells are presented. This radius has been optimized on REV? What is the sensitivity to this parameter? | The radius was not specifically optimised on REV, the selected radius was based on the previous article (Courdent et al., 2016), which describes this method further. We will make this clearer. |
| 3. P5 L14 UTC | Yes, UTC will be added to the NWP generation time. |
| 4. P6 L18 and other occurrences of "forecasted" should be "forecast". "summing up to a total of n event assessments" : this part of the description methodology should be made much clearer (see general comments). | The occurrences of "forecasted" will be corrected to "forecast".

We will clarify the computation of the skill score as mentioned in the reply to the general comment [3]. |
| 5. P7 L4 "Methodology" : section 2 involves also description of the methodology. Section 3 is more related with validation. | This comment is similar to the reviewer 1 specific comment [8].

The NWP post-processing methods described in section 2.2 are developed in the previous manuscript (Courdent et al. 2016) and are used to generate input data to the model for this article.

Despite being a method, it was decided to develop it under the data section to distinguish it from the core methods of this paper. This section will be modified to make this clearer. |
| 6. P10 L6 and other occurrences (e.g. Table 5, Figure 5, 8) "always optimize", "never optimize" sounds strange because the paper is all about optimization. Find a short reference to the two objectives (like "always energy objective"). ½½½ | We agree the text on the figure a will be changed for more clarity. |
| 7. P11 L20-29, Figure 7b, Table 4 The slightly better upper bound provided by the areal overlap method can't be seen on the figure. How the complementarity of both post-processing approaches can be used in a real situation? Through the gain/loss ratio and the decision threshold? | The complementarity of the 2 approaches is more visible on the ROC diagram Figure 7a. The catchment aerial overlap provides valuable information for low PoFD, which are not covered by Maximal threat approach.

This is not especially visible on the REV diagram |

1 | P a g e

**Fig. 1.**

---

## Author Comment (AC3) · 18 Jan 2017

Dear reviewer,

We greatly appreciate the review and acknowledge that the comments and suggestions will lead to an improved paper. Our reply to the general comments:

I can find little to fault the paper. My main query would be that I could find no details or reference as to how the system is operated during the 'optimised' phase apart from

relatively vague statements. While the paper is interesting without this, it does limit the understanding / reproducibility. I can see that the omission of this may be due to commercial reasons given the authors' affiliations, but it would be useful to explicitly state this.[1]

[1] The aim of this article is to provide a framework using EPS NWP to predict when a management switch to IUWDS is beneficial considering the uncertainty of the weather forecast. The control strategy needed for the optimisation scheme is developed in another manuscript by R. Halvgaard et al. that is currently in review (see below).

R. Halvgaard, L. Vezzaro, P. S. Mikkelsen, M. Grum, T. Munk-Nielsen, P. Tychsen, H. Madsen: "Intergrated Model Predictive Control of Wastewater Treatment Plants and Sewer Systems in a Smart Grid" (in Review Process).

It would also be useful to understand more details of the calibration of the hydrological model even if just a few short sentences. [2]

[2] This comment is similar to the general comment [1] and the specific comment [10] from the first reviewer.

Section 2.3 (P5 and 6) will be split in 2, the first on the study case which will be expanded with more data on the WWTP and the second on the hydrological model which will be further developed. We will also explain with more clarity that further information on the hydrological model is available in (Courdent et al., 2016).

While clearly outside the scope of this paper, it would be of great interest to see any results from a real world implementation of the proposed framework should it be implemented! [3]

[3] We agree that further information on results and performance would be appreciated and we are working towards it. Indeed, two large pipes will be constructed just before the inlet to the Damhusåen WWTP with the primary purpose to reduce CSO to cope with new regulations. Those 2 pipes can contained a volume corresponding to one day

of dry weather flow and would nicely fit the concept developed in this paper and in R. Halvgaard et al. paper.

We will make changes to the paper that accommodate the technical comments by the reviewer, including careful proofreading. We would like to express our sincere thanks to the reviewers for their constructive comments and identification of areas in the manuscript which needed clarification.

On behalf of all the authors,

Vianney Courdent
* * *

---

## Author Response (AR1)

Dear reviewers,

This combined pdf file gathers the point-by-point response to the 3 reviews and the marked-up revised manuscript as display in the table of contents below.

Accordingly to our response to the first specific comment of reviewer 1, the title of the manuscript was changed to:

"A gain-loss framework based on ensemble flow forecasts to switch the urban drainage-wastewater system management towards energy optimization during dry periods."

**Table of Contents**

We would like to express our sincere thanks to the reviewers for their constructive comments and identification of areas in the manuscript which needed clarification.

On behalf of all the authors,

Vianney Courdent

Dear reviewer,

We greatly appreciate the review and acknowledge that the comments and suggestions will lead to an improved paper.

Please find in the table below our point-by-point response to the comments.

| Referee # 1 | |
|---|---|
| **General comments** | |
| *The manuscript presents an interesting topic within the scope of the journal. While enough information on the used methodology is given to enable a rough understanding, the reader will not be able to reproduce the applied methods based on that information. Result interpretation will be difficult for the reader without more specific information on the used hydrological model (its calibration and accuracy) [1] and the WWTP and the potential impact of the described method on its performance in terms of energy consumption and effluent water quality [2].* | [1] Section 2.3 (P5&6) was split in 2, the first on the study case was expended with more data on the WWTP and the second on the hydrological model was further developed. The relation with the previous article (Courdent et al., 2016) which further described the model was made clearer.

As suggested in the specific comment 19, the model output based on rain gauge data input will be added on figure 9 to illustrate the performance of the hydrological model.

Courdent, V., Grum, M. and Mikkelsen, P. S.: Distinguishing high and low flow domains in urban drainage systems 2 days ahead using numerical weather prediction ensembles, J. Hydrol., doi:http://dx.doi.org/10.1016/j.jhydrol.2016.08.015, 2016. |
| | [2] Further information on the WWTP energy consumption was added in the study case section (P4 L34ff) and additional information was added to the results and discussions section (P9 L37ff).

We do not have precise data regarding the impact on the effluent water quality. However the effluent water quality is of primary interest. The Danish WWTPs pay taxes on their effluent pollution load. Therefore the boundaries of the energy optimization scheme will be defined to respect the performance of the WWTP.

Halvgaard et al. (submitted 2017), presents a control method that takes the effluent quality (nitrogen) into account. A reference to this article was added to the outlook section (P10 L23ff).

R. Halvgaard, L. Vezzaro, P. S. Mikkelsen, M. Grum, T. Munk-Nielsen, P. Tychsen, H. Madsen: Integrated Model Predictive Control of Wastewater Treatment Plants and Sewer Systems in a Smart Grid, Computer |

| | & Chemical Engineering, submitted 2017. |
|---|---|
| *More information in the results section on the performance of the proposed method and a discussion of its risks and benefits would be very desirable as without them it is difficult to draw clear conclusions about the value of the method [3].* | [3] The scope of this article is to develop a method to determine when the IUDWS management can be switched to optimizing its energy consumption, which is possible during dry weather (flow below a given threshold).

As mentioned in the updated outlook section (P10 L23ff), Halvgaard et al. (submitted 2017) presents a Model Predictive Control (MPC) to control the power consumption of pumps in a sewer system and the treatment power consumption according to electricity prices and effluent quality (nitrogen). This article is currently under review for the Computer & Chemical Engineering journal.

We agree that further information on results and performance would be appreciated and we are currently working towards it. As mentioned in the updated outlook section (P10 L 19), two large pipes will be constructed just before the inlet to the Damhusåen WWTP with the primary purpose to reduce CSO to cope with new regulations. These 2 pipes can contain a volume corresponding to one day of dry weather flow and would nicely fit the concept developed in this paper and in the paper by R. Halvgaard et al. (submitted 2017) |

| **Specific comments** | |
|---|---|
| 1. P1 L1: This is one long title. Would "Coupling urban drainage-wastewater systems and electric smart grids during dry periods" not suffice? Even then I find the title somewhat misleading: Is the WWTP not continuously coupled to the smart grid? If I understand correctly, rather than coupling and uncoupling, only the temporal amount of energy consumption is optimized depending on the hydraulic condition. | The title was changed to :

"A gain-loss framework based on ensemble flow forecasts to switch the urban drainage-wastewater system management towards energy optimization during dry periods." |
| 2. P2 L8ff: Maybe I misunderstand this sentence, but it seems to suggest that only because rain only occurs 7 % of the time it makes sense to look into energy optimization. Would it make less sense if you had 10 % of rain? Could you clarify/rephrase this? | P2 L8ff The sentence was modified for more clarity. |
| 3. P2 L8ff: Dry weather flow rarely can be defined by | |

| | |
|---|---|
| "no rain", usually (sufficient sewer network size and event intensity) wet weather conditions will be predominant for several hours after a rain event has ended (as the proposed method does by using the flow rather than the rain as a trigger for switching between control objectives). I suggest rephrasing this section accordingly. | |
| 4. P2 L10ff (same phrase again, sorry): Striving for energy optimization and emission reduction of WWTP is standard practice for many years – both during dry and wet weather. With this background in mind it would be better to slightly rephrase and cite some relevant literature here to avoid this phrase being interpreted as a novel suggestion as such. | |
| 5. P3 L15ff: Some details are discussed here that re-occur in section 3. Delete here? | As suggested this part was deleted to avoid reoccurrence. |
| 6. P4 L5: A clear(er) definition of REV (as eg in the abstract) would be critical here as it forms the basis of the paper | P4 L2ff A sentence was added to describe the REV with more clarity. |
| 7. P5 L20ff: Are methods (i) and (ii) defined in literature? They seem to be a mix of other methods. It would not be possible to reproduce your method from this section. Please give more specific information on the used methods for enhancing the forecast. | P5 L17: The reference to the just published manuscript (Courdent et al. 2016) was made clearer. |
| 8. P6 section 2 and especially 2.2 discusses methods. I'd suggest to move these parts to section 3 | The NWP post-processing methods described in section 2.2 are developed in the previous manuscript (Courdent et al. 2016) and are used to generate input data to the model for this article.

It was decided to develop it under the data section to distinguish it from the core methods of this paper. |
| 9. P6 L20ff: You mention that the DWF module of the model is calibrated, but there are no details on the calibration of the Nash cascades. These details seem essential for accurate predictions, please add them to this section. | P6 L20ff As mentioned in the general comments [1] this section was be split in 2 to further describe the hydrological model, and the link to the previous article (Courdent et al. 2016) developing the hydrological model was made clearer. |
| 10. P5, section 2.3: No details on the WWTP and its energy consumption are given while this forms the focus of the paper [1]. Changed flow regimes at the influent will necessarily cause changes in energy consumption of the WWTP. How are these considered | [1]As mentioned in the reply to general comments, section 2.3 was be split in 2 and further details on the WWTP were added in the study case section (P6 L1).

[2] Indeed, the control of the energy consumption |

| in this study? [2] | based on the energy market can result in a decrease of the cost together with an increase of the energy consumption as explained in the added reference (Aymerich et al., 2015) in P12 L23ff.

This aspect is out of the scope of this article. We added in the outlook section (P13 L11ff ) a reference to (Halvgaard et al., submitted 2017) which is currently under review. This article presents a Model Predictive Control (MPC) to control the power consumption of pumps in a sewer system and the treatment power consumption according to electricity prices

Aymerich, I., Rieger, L., Sobhani, R., Rosso, D., Corominas, L., 2015. The difference between energy consumption and energy cost: Modelling energy tariff structures for water resource recovery facilities. Water Res. 81, 113–123. doi:10.1016/j.watres.2015.04.033 |
| --- | --- |
| 11. P6 L31ff: Move to introduction? | This part of the section 2.4 on energy market section was moved to the introduction (P2 L17ff). |
| 12. P7 L5ff: Leave out? Does not add anything to the understanding of the reader. | The introduction to the methodology section was removed. |
| 13. P11 L5ff: are these not results that should be moved to the next section? | This part was moved to the result section (P11 L3ff). |
| 14. Figure 8 is not interpreted in the text. Omit? | Figure 8 is mentioned in P12 L9. |
| 15. P12 L13: How big are the uncertainties in the energy price forecasts? Could you comment on their (potential) influence on your method? | The term "forecasted" was changed for more clarity (P12 L10).

The energy price for the incoming day is set through the energy market (Nord pool for Denmark) based on bids and offers and is therefore fixed without uncertainty.

"Buyers and suppliers submit bids and offers for each hour of the next day and each hourly MCP (market clearing price) is set such that it balances supply and demand." (Weron 2006)

The smart grid section was reshaped, part of it was moved to the introduction (see reply to comment 11) and additional information on the electricity market were added (there are different electricity markets with different lead times, e.g. the day-ahead market have 24 |

| | hours lead time whereas the intraday market has 1 hour lead time. Bids and offers made on the first market can be adapted on the second). |
| --- | --- |
| | Weron, R.: Modeling and forecasting electricity loads and prices: A statistical approach, First Edit., John Wiley & Sons Ltd., 2006. |
| 16. P12 L15ff: In this scenario, it is possible to save 26 €MWh during 2 days. I suggest to add information that answers at least a number of the following questions in order for the reader to be able to understand the meaning of this result: Is this a representative result? What is the average/peak energy consumption of the WWTP? What is the maximum amount of energy that could be switched? How much could be saved during the 2 years of data you used? How does that impact the total energy consumption of the WWTP ('switching' could result in an increase as well as a decrease)? What is the influence on the WWTP effluent performance and emission of greenhouse gasses? | This comment is similar to general comments [2]. Further information on the WWTP energy consumption (e.g. energy consumption per $m^3$) was added to allow the reader to have a better understanding of the meaning of this result (P6 L1ff and P12 L15ff). As mentioned in the reply to comment 10 and 18, energy optimization based on the energy price can result in an increase of the total energy consumption. A references was added to underline this possibility (P12 L23). The energy consumption optimisation scheme (not developed in this article) has to include the WWTP performance within its decision criteria. E.g. (R. Halvgaard et al., submitted 2017) mentioned in the outlook (P13 L11ff) used the nitrogen concentration as a measure of effluent quality. The impact on emission of greenhouse gasses was not directly assessed. However, daily peaks in waste water usually coincide with peak demand on the power grid, thus coinciding with the highest energy price periods. Hence, reducing these wastewater inflow peaks when energy cost are high will also benefit the energy system by reducing grid load and GHG emissions (due to the need for more carbon- intensive energy sources during peak power demand periods). |
| 17. P12 L15ff: What is the cost of the suggested system (at the least the NWP data will have to be purchased + some man-hours for keeping the real-time system up and running) as compared to its benefits? It seems these considerations should be included in order to judge the actual gains produced by the system. | A large project is currently under implementation on the downstream part of the study case catchment just before the inlet to the WWTP. Two large pipes with a storage volume corresponding to the daily dry weather flow to the WWTP will be constructed to cope with new regulations on CSO. The volume of these two pipes would nicely fir the concept developed in this article (P13 L7) and provide further information on the cost of the system. A reference to this project was added to the outlook |

| | section (P10 L19ff) |
|---|---|
| 18. P12 L24 "optimization based on economic objectives will also yield environmental benefits": This seems a much too broad statement that should be explained or based on a citation. In the context of this paper, it seems that it would be perfectly possible to create a scenario where 'switching' energy consumption would lead to an overall increase of energy consumption (e.g. by running the blowers on a frequency at which they are less efficient than when not 'switching'), but a decrease in cost. | The intended message of this sentence is that the correlation between energy price and proportion of wind energy leads to the consumption of energy with a lower CO2 footprint. But indeed as you rightfully pointed out the optimisation can result in an increase of the overall energy consumption. The sentence was motived to clarify this point (P12 L22ff). |
| 19. P12 L26: "most EMs predict the high flow event but often too early" Is this a problem of the precipitation prediction or the hydrological model? This would be difficult for the reader to judge (without information on the calibration of the hydrological model). Maybe you could add a line in figure 9: the output of the hydrological model given the observed rain. This would also (more or less) address the above comment for P6 L20ff. | Figure 9 was updated accordingly. |
| Technical corrections | Thank you for the technical corrections, which are accommodated in this revised version of the manuscript. |

Dear reviewer,

We greatly appreciate the review and acknowledge that the comments and suggestions will lead to an improved paper.

Please find in the table below our point-by-point response to the comments.

| Referee # 2 | |
|---|---|
| **General comments** | |
| *The problem is exposed as a dynamic one (e.g. P4 L25-26, P9 L27-30, P14 L21-22).However it is solved as a static one: the relative economic value is presented as a function of the gain/loss ratio [1]. The decision threshold (i.e. the fraction of members of the ensemble of predicted discharge exceeding a discharge threshold) beyond which the manager decides to switch from energy optimization to safety of the system is deduced from the envelope of separate curves. As for the choice of a method for the post-processing of numerical weather prediction model the maximum threat method extends significantly the range towards low gain/loss ratios resulting in positive relative economic values compared with the aerial overlap method. Having these results, the methodology has still to be proven in real dynamic situation i.e. where the decision to be taken at a given time depends on the decisions already taken. What is missing in order to that? An order of magnitude of the losses in case of combined sewer overflow, a hydraulic model able to reflect the management actions? The authors are asked to make clear the scope of the paper and either add new results or add comments in the outlooks [2].* | [1] Thank you to point this out, you are right. The use of the term "dynamic" was corrected for more clarity.

The referenced articles on the REV (Richardson, 2000; Roulin, 2006) are using fixed alpha values (the cost of mitigation measures and their benefits are assumed fixed). In our case the alpha value varies with time, the gain depends directly on the variation of the energy market. Hence the incentive to optimize the IUDWS for energy consumption changes with time. In this sense the alpha ratio is dynamic. But, indeed for at a given time, for given NWP forecast the alpha is fixed and the problem solved as a static one. |
| | [2] We agree that further information on results and performance would be appreciated and we are working towards it. As mentioned in the updated outlook section (P10 L 19), two large pipes will be constructed just before the inlet to the Damhusåen WWTP with the primary purpose to reduce CSO to cope with the new CSO regulations. Those 2 pipes can contain a volume corresponding to one day of dry weather flow and would nicely fit the concept developed in this paper and in the paper by R. Halvgaard et al.

R. Halvgaard, L. Vezzaro, P. S. Mikkelsen, M. Grum, T. Munk-Nielsen, P. Tychsen, H. Madsen: Integrated Model Predictive Control of Wastewater Treatment Plants and Sewer Systems in a Smart Grid, Computer & Chemical Engineering, submitted 2017. |
| *I missed information regarding the methodology. No lead time is specified with the results. Are all the ensembles (2 years x 4 issue hours) used at hourly time step to the forecast horizon of 54 hours (P5 L14-18)? Or 2 days (P3 L5, P12 L11-26)? How the scores are computed regarding both the issue time and the lead* | [3] Indeed a decrease in the performance skill is observed with increasing lead time as was documented in (Courdent, 2016). The bids and offers on the daily energy market are made up to 36 hours in advance. Therefore, the different lead times are aggregated in the results. This point was added in the methodology |

| | |
|---|---|
| *time? In case the forecast horizon is 2 days, how do the authors deal with the decreasing skill scores or relative economic value of the predictions with the lead time? [3]* | section (P6 L2ff). |

| **Specific comments** | |
|---|---|
| 1. P3 L26 "spatial" instead of "special". Do you have a reference for this assertion? | Indeed it is a typo, we meant "spatial". This misspelling was corrected. |
| 2. P4 L1 "The radius of the neighbouring area included is used as a parameter during the decision making, in addition to fEM." should move from introduction to the methodology section (2 Material: : : : 2.2). P11 L18, Figure 7 Results considering a radius of 6 grid cells are presented. This radius has been optimized on REV? What is the sensitivity to this parameter? | The radius was not specifically optimised on REV, the selected radius was based on the previous article (Courdent et al., 2016), which describes this method further. |
| 3. P5 L14 UTC | UTC was added to the NWP generation time. |
| 4. P6 L18 and other occurrences of "forecasted" should be "forecast". "summing up to a total of n event assessments" : this part of the description methodology should be made much clearer (see general comments). | The occurrences of "forecasted" were corrected to "forecast".

As mention in the reply to the general comment [3], the description of an event was made clearer in the methodology section (P6 L2ff). |
| 5. P7 L4 "Methodology" : section 2 involves also description of the methodology. Section 3 is more related with validation. | This comment is similar to the reviewer 1 specific comment [8].

The NWP post-processing methods described in section 2.2 are developed in the previous manuscript (Courdent et al. 2016) and are used to generate input data to the model for this article.

Despite being a method, it was decided to develop it under the data section to distinguish it from the core methods of this paper. This section was modified to make this clearer. |
| 6. P10 L6 and other occurrences (e.g. Table 5, Figure 5, 8) "always optimize", "never optimize" sounds strange because the paper is all about optimization. Find a short reference to the two objectives (like "always energy objective"). ½½ | We agree the text on the figure a will be changed for more clarity. |

| | |
|---|---|
| 7. P11 L20-29, Figure 7b, Table 4 The slightly better upper bound provided by the areal overlap method can't be seen on the figure. How the complementarity of both post-processing approaches can be used in a real situation? Through the gain/loss ratio and the decision threshold? | The complementarity of the 2 approaches is more visible on the ROC diagram Figure 7a. The catchment areal overlap provides valuable information for low PoFD, which are not covered by Maximal threat approach.

This is not especially visible on the REV diagram Figure 7b due to the logarithmic scale and the allocated value to the different outcomes (Table 3) (i.e. no actions are taken in case of positive forecast and therefore no value is allocated to hits and false).

With another definition of decision making, the complementary could be more pronounced. E.g. REV on mitigation measures (e.g. flood prediction) will allocate the values differently and false alarms may have a high impact. |
| 8. P12 L15-18 "can yield up to 8 €" what is the order of energy consumption we are dealing with? What is missing is an overall estimation of the cost of energy during the 2 years and how much is gained during the same period using the switching strategy optimized based on REV results. | This comment is similar to the specific comment [16] of Reviewer 1.

Further information on the WWTP energy consumption (e.g. energy consumption per $m^3$) was added in the case study section (P4 L34ff) and in the results section (P9 L37ff).

The aim of this article is to provide a framework using EPS NWP to predict when a management switch to IUWDS is beneficial considering the uncertainty of the weather forecast. As mentioned in the updated outlook section (P10 L23ff), the control strategy needed for the optimisation scheme is developed in another manuscript by R. Halvgaard et al. currently under review.

R. Halvgaard, L. Vezzaro, P. S. Mikkelsen, M. Grum, T. Munk-Nielsen, P. Tychsen, H. Madsen: Integrated Model Predictive Control of Wastewater Treatment Plants and Sewer Systems in a Smart Grid, Computer & Chemical Engineering, submitted 2017. |
| 9. P24 Figure 9(a) Add a legend for the two curves. | The meanings of the two curves are described by the axis label and colour, a legend was added for more clarity. |

Dear reviewer,

We greatly appreciate the review and acknowledge that the comments and suggestions will lead to an improved paper.

Please find in the table below our point-by-point response to the comments.

| Referee # 3 | |
| --- | --- |
| **General comments** | |
| *I can find little to fault the paper. My main query would be that I could find no details or reference as to how the system is operated during the 'optimised' phase apart from relatively vague statements. While the paper is interesting without this, it does limit the understanding / reproducibility. I can see that the omission of this may be due to commercial reasons given the authors' affiliations, but it would be useful to explicitly state this.[1]* | [1] The aim of this article is to provide a framework using EPS NWP to predict when a management switch to IUWDS is beneficial considering the uncertainty of the weather forecast. As mentioned in the updated outlook section (P10 L23ff), the control strategy of an optimisation scheme is developed in another manuscript by R. Halvgaard et al. currently under review for the Computer & Chemical Engineering journal.

R. Halvgaard, L. Vezzaro, P. S. Mikkelsen, M. Grum, T. Munk-Nielsen, P. Tychsen, H. Madsen: Integrated Model Predictive Control of Wastewater Treatment Plants and Sewer Systems in a Smart Grid, Computer & Chemical Engineering , submitted 2017. |
| *It would also be useful to understand more details of the calibration of the hydrological model even if just a few short sentences. [2]* | [2] This comment is similar to the general comment [1] and the specific comment [10] from the first reviewer.

[1] Section 2.3 (P5&6) was split in 2, the first on the study case was expended with more data on the WWTP and the second on the hydrological model was further developed. The relation with the previous article (Courdent et al., 2016) which further described the model was made clearer.

Courdent, V., Grum, M. and Mikkelsen, P. S.: Distinguishing high and low flow domains in urban drainage systems 2 days ahead using numerical weather prediction ensembles, J. Hydrol., doi:http://dx.doi.org/10.1016/j.jhydrol.2016.08.015, 2016. |
| *While clearly outside the scope of this paper, it would be of great interest to see any results from a real world implementation of the proposed framework should it be implemented! [3]* | [3] We agree that further information on results and performance would be appreciated and we are working towards it. As mentioned in the updated outlook section (P10 L 19), two large pipes will be constructed just before the inlet to the Damhusåen WWTP with the |

| | primary purpose to reduce CSO to cope with the new CSO regulations. Those 2 pipes can contain a volume corresponding to one day of dry weather flow and would nicely fit the concept developed in this paper and in the paper by (R. Halvgaard et al., submitted 2017) |
|---|---|

[revised manuscript text omitted]

**Comment [VATC1]:** New figure – "Never optimized" replaced by "Never energy objective"
"Always optimized" replaced by "Always energy objective"

[Figure]

**Figure 6: ROC and REV diagram for flow domain forecast based on catchment weighted areal overlap.**

Comment [VATC2]: New figure.

Correction in the legend, change in location of the number (3, 4, 5).

[Figure]

Comment [VATC3]: New figure.

Correct typo in legend

**Figure 7: ROC and REV diagram for flow forecasts considering the two NWP post-processing methods: The maximal threat EPS method with a neighbourhood radius of 6 grid cells in colour and the catchment weighted areal overlap method in grey colour as background.**

[Figure]

Comment [VATC4]: New figure

"Never optimized" replaced by "Never energy objective"
"Always optimized" replaced by "Always energy objective"

**Figure 8:** *REV* curves for the EPS NWP post-processing maximal threat in a radius of 6 grid cells from the catchment (a, left plot) and best decision threshold according to the α-value (b, right plot), in blue for the maximal threshold approach and in grey for the areal overlap approach.

[Figure]

Comment [VATC5]: New figure with legend in (a) and modelled discharge from rainfall measurement.

**Figure 9:** Example illustration of the EPS flow prediction system for two selected days, 29-30 April 2015. Energy market parameters, energy price and proportion of wind power (1, top plot), ensemble flow predictions using the areal average (b) and maximal threat (c) post-processing methods, and (d) flow domain predictions for the two post-processing methods and for each two decisions thresholds, cf. Table 5 (coloured areas imply that an event is predicted, otherwise not).